# Do residential patterns affect women's labor market performance? An empirical study based on CHFS data

Siyan Zhou *, Qing Wang

Institute of Chinese Financial Studies, Southwestern University of Finance and Economics, Chengdu City, Sichuan Province, China

¤ Current address: Southwestern University of Finance and Economics, Wenjiang District, Chengdu City, Sichuan Province, China
* 357816494@qq.com

**Data Availability Statement:** The data underlying the results presented in the study are available from the Survey and Research center for China Household Finance. If readers want to use the data, they can visit the website: https://chfs.swufe.edu.

## Abstract

Based on China Household Finance Survey (CHFS) data from 2019, this paper explores the impact of the residential pattern of coresidence with parents on the labor market performance of women in married families with minor children. The study finds that coresidence with parents significantly increases the possibility of female labor market participation and positively impacts women's employment income. To overcome the potential endogeneity problem of residential patterns, this paper uses the Heckman two-step method and the conditional mixed process estimation method (CMP method) for regression, and the conclusions remain robust. The mechanism analysis shows that coresidence with parents has both grandchild care and elderly care factors, which have a spillover effect and a crowding-out effect on female labor market performance, respectively. Since the spillover effect is more significant than the crowding-out effect, coresidence with parents positively impacts women's labor market performance. The heterogeneity analysis shows that in terms of labor force participation rate, coresidence with parents has a more significant impact on women in families with children aged 0–6, women in families without boys, and women in families with employed husbands. In terms of income, coresidence with parents has a more significant impact on women in families with employed husbands. This study provides a new perspective for promoting female labor market performance and can serve as a reference for future policy formulation.

## 1. Introduction

China's female labor participation rate has long been the focus of academic attention. Compared with the world average, China's female labor force participation rate is still high; however, in recent years, China's labor market participation rate has gradually declined, with the female labor force participation rate declining more rapidly than that of men. Women have also been lagging in terms of labor force participation and income compared with men. Based on 2011–2019 survey data from the Family Financial Center of the Southwest University of

cn. We have also uploaded the data to the public database OPENICPSR, Resource URI: /openicpsr/194102/. Please visit the following URL to access the project: https://deposit.icpsr.umich.edu/deposit/claimResource?tenant=openicpsr&claimId=120703.

**Funding:** This study was funded by National Natural Science Foundation of China (NSFC), Grant number: 7194100222. The funders had no role in study design, data collection and analysis, decision to publish, or preparation of the manuscript. The funders provided only financial support for this study. There was no additional external funding received for this study.

**Competing interests:** The authors have declared that no competing interests exist.

Finance and Economics, this paper selects married families as samples, which are plotted Figs 1 and 2. The sample age of married women is controlled to between 20 and 55 years old, while the sample age of married men is controlled to between 22 and 60 years old. Families with children are retained as samples only if their children are 18 years old or younger. According to Figs 1 and 2, married men's labor force participation rate and income have been significantly higher than those of married women over the years. The labor force participation rate and income of married and fertile men are also significantly higher than those of married and fertile women, with the gap between married and fertile groups being more obvious.

The female labor force is an important and special labor force in the market. It is important because women basically occupy half of the labor market, and it is special because traditional social concepts and invisible discrimination have a subtle influence on women, thus hindering their performance in the labor market [1]. With the gradual development of the modern economy and changes in the mode of labor and division of labor in society, the modern concept of gender equality has gradually become the mainstream consciousness; however, the reality of discrimination against women still exists, and women are more or less affected by traditional role concepts, such as "men are responsible for the outside world, while women are responsible for the inside world" [2, 3]. Under the influence of traditional gender concepts, married women usually devote more time and energy to nonmarket work such as supporting parents, taking care of children, and supporting their spouses in gainful employment, which correspondingly reduces their labor market supply and human capital accumulation [4–6]. Conversely, men are more active in the labor market and strive to obtain more remuneration [7]. In addition, if a large part of society has a negative view of women in working in paid positions, then the effect of women's choice of paid work will be reduced because of society's view, which will in turn reduce the probability of women working in their choice of paid positions. Therefore, traditional cultural concepts and invisible discrimination against women in a society that is under the influence of traditional cultural concepts have a negative impact on women's labor market performance; this is one of the most important reasons that women's labor market participation rate is lower than that of men in the long term [8–12]. With the gradual decline in China's demographic dividend, it is of great practical significance to improve the labor participation rate and income level of women against the background that the labor participation rate and income level of married and fertile men are significantly higher than those of women.

Under the influence of traditional cultural concepts, women who are married and have children tend to bear more family care-related pressure than men. Since childcare-related stress tends to be concentrated in the stage of caring for minor children, it is increasingly common for couples to live with their parents during the early years of a child's life [13]. However, there is little research on this phenomenon in the economic literature. This residential pattern directly affects the amount of time and energy women spend on family caregiving, which in turn affects women's labor market performance. For women who are married and have minor children, living with elderly relatives may have both a crowding-out effect and a spillover effect, which will produce an opposite effect on their labor market performance. On the one hand, living with elderly relatives may increase the level of elderly care-related pressure placed on married women, resulting in a crowding-out effect; on the other hand, elderly relatives can help married women take care of children and ease women's parenting-related stress, resulting in a spillover effect. Thus, living with elderly relatives affects women's labor market performance due to the combination of these two effects. However, most of the recent literature focuses only on the negative impact of elderly care on women's employment or the promotion of grandchild care on women's employment; little literature focuses on the two effects at the same time and sorts out the specific impacts and internal mechanisms of the residential pattern of coresidence with parents on women's labor market performance. In addition, most of

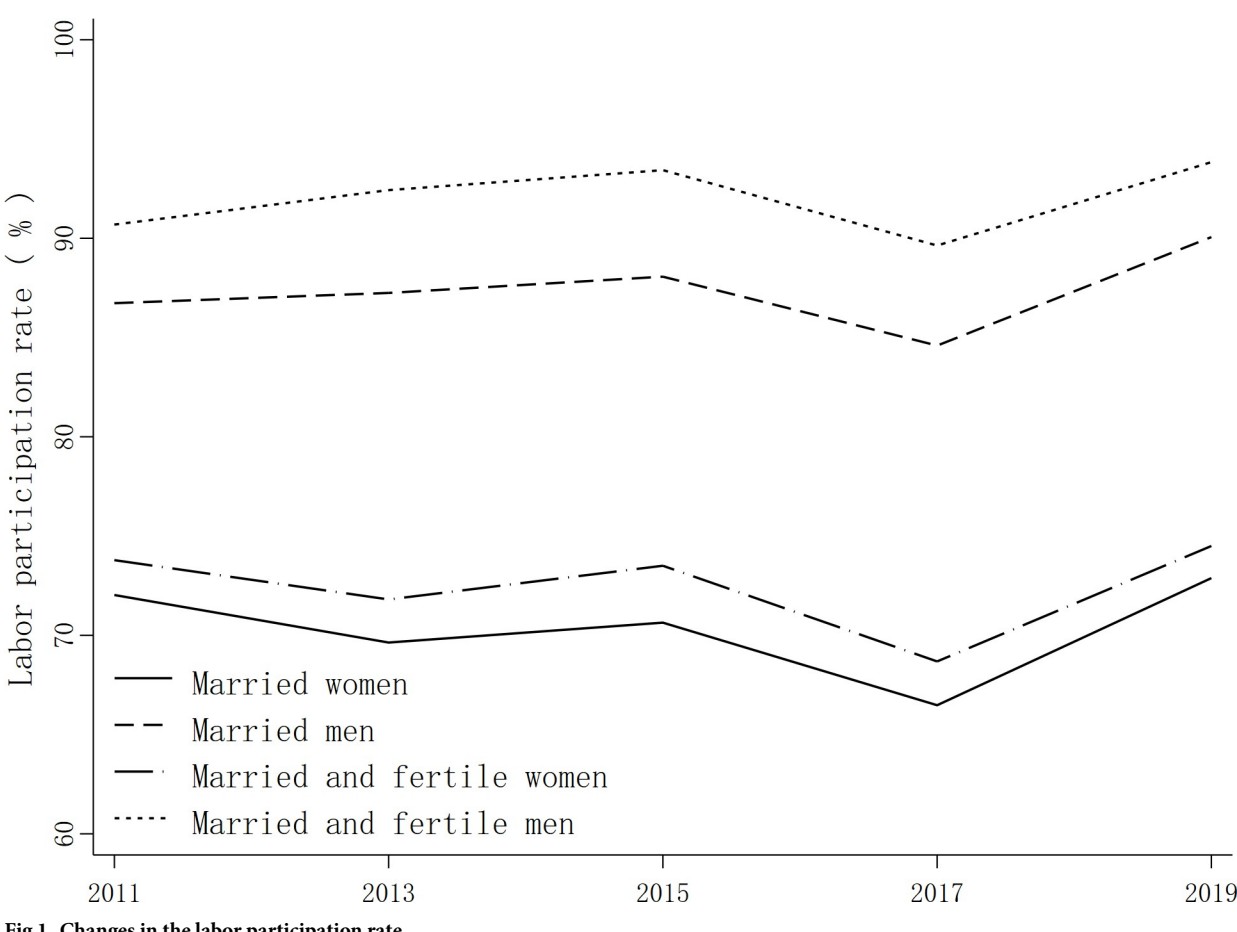

**Fig 1. Changes in the labor participation rate.**

the literature focuses on only the impact of elderly care or grandchild care on women's labor market participation and does not further discuss its impact on women's income. Based on this, this paper uses 2019 China Household Finance Survey (CHFS) data and adopts the method of empirical research to explore the specific impact of living with elderly parents on the labor market participation and income of women in married families with minor children and analyze its internal mechanisms.

The results of this paper show that, first, for women who are married and have minor children, living with parents significantly promotes their labor market participation possibilities and has a positive impact on their income. Second, in terms of the labor force participation rate, the examined residential pattern has a more significant effect on women in families with children aged 0–6, women in families without boys, and women in families with employed husbands. In terms of income, the examined residential pattern has a more significant impact on women in families with employed husbands. Third, there is grandchild care associated with living with parents, which has a spillover effect on female labor market performance; there is also elderly care associated with living with parents, which has a crowding-out effect on female labor market performance. Since the spillover effect is more significant than the crowding-out effect, living with parents has a positive impact on women's labor market performance.

Compared with the existing research, this paper may contribute to the following aspects. First, previous studies have mainly focused on the negative effects of elderly care or the positive effects of grandchild care; however, it is likely that both type of effects are commonly present.

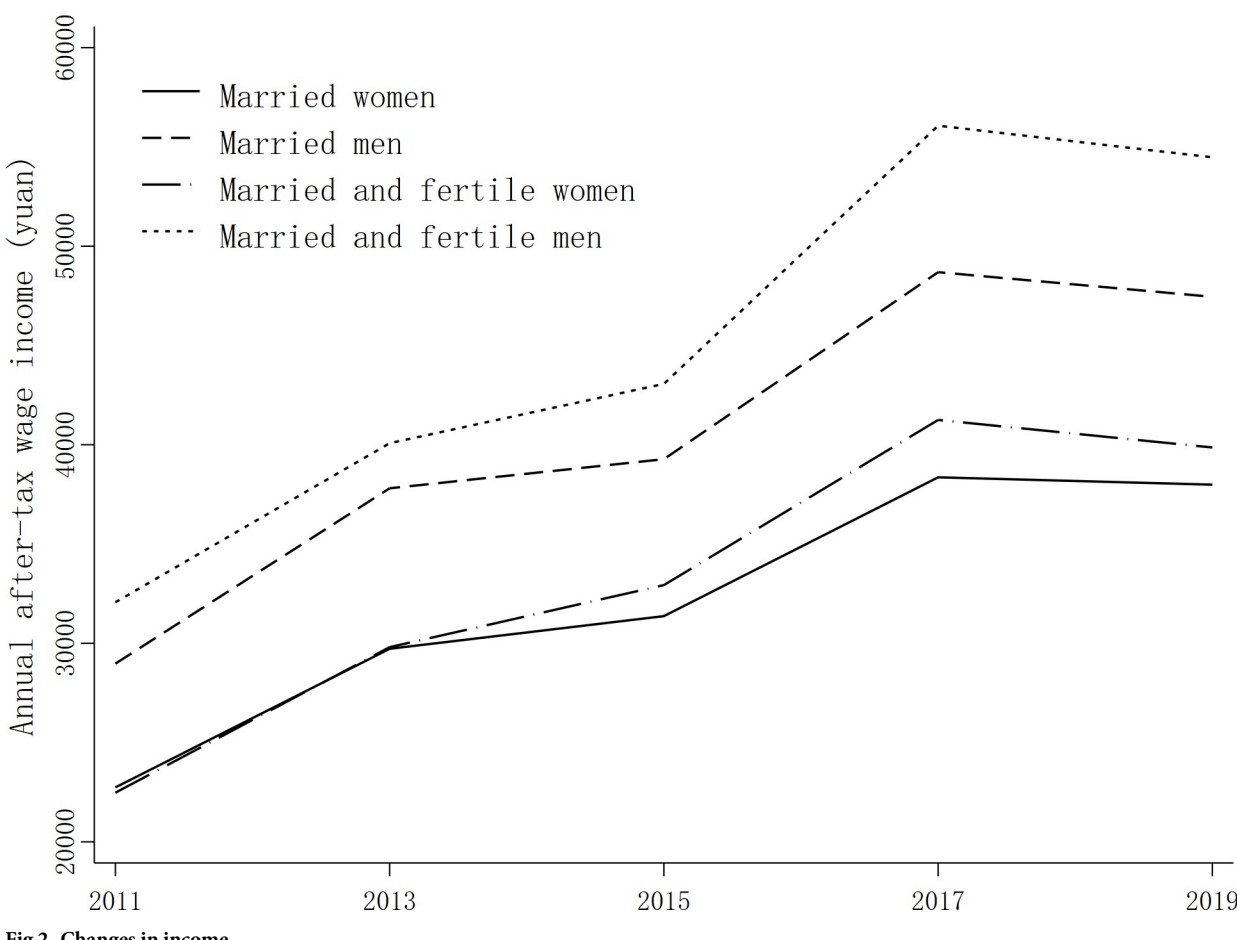

**Fig 2. Changes in income.**

Based on this, the current paper discusses the impact of residential patterns on women's labor market participation and income. Then, it discusses the specific mechanisms based on perspectives of a spillover effect and a crowding-out effect, which provides a more comprehensive perspective. Second, this paper further analyzes the heterogeneous effects of residential patterns on women in different families based on the different age groups of children, the presence of boys, and whether husbands are employed, making relevant policy recommendations more targeted. Third, this paper takes the ratio of families living with their parents in the county where the family is located (this family is not included in the sample calculation) and whether the head of household has the concept of "raising children for old-age support" as the instrumental variable and draws on the conditional mixed process estimation method proposed by Roodman [14] for regression, thereby effectively overcoming the potential endogenous bias in the analysis process and resulting in more robust conclusions. Women are an important source of labor supply in the labor market. Against the background of the acceleration of the aging process in China's population, the gradual disappearance of the demographic dividend, and the problem of insufficient labor supply, this paper discusses the impact of the residential pattern of coresidence with parents on the labor market performance of women in married families with minor children; the findings will enrich the relevant research and serve as points of reference for relevant departments to formulate policies that promote female employment.

The rest of this paper is expanded according to the following ideas. The second part consists of the theoretical basis and research hypothesis, and the third part consists of the research

method. The fourth part consists of the empirical analysis, including benchmark regression and mechanism test. The fifth part consists of further discussion, including endogenous problem discussion, heterogeneity analysis and robustness testing. The sixth part is discussion, including relevant policy recommendations, limitations of this paper and future research direction. The last part is the conclusion of the whole paper.

## 2. Theoretical basis and research hypothesis

Women are an essential driving force for societal progress and national economic development. With the advancement of women's social status, the issue of female employment has garnered significant attention from scholars both domestically and internationally. Compared to men, the factors influencing women's employment are more numerous and complex, including family income, educational attainment, and fertility policies, among others. Scholars have conducted extensive research from various angles. Research based on the human capital theory primarily examines the roles of individual characteristics, such as age, household registration, education level, marital status, work experience, and health status, in influencing female employment [15, 16]. Studies grounded in the family economic theory focus on the impact of characteristics like husband's income, number of children, parental care, allocation of household time, and family economic conditions [17, 18]. Some scholars explore the impact mechanisms of macroeconomic factors such as economic transition, foreign trade, healthcare, and pension insurance systems on female employment [19]. Some researchers conduct comparative studies based on micro samples from multiple countries to investigate factors affecting women's participation in the labor market, revealing that in relatively impoverished countries, a substantial portion of women work out of economic necessity [20].

However, previous literature on the factors influencing female employment has often taken a somewhat narrow perspective. In China, the phenomenon of multiple generations living under the same roof is becoming increasingly common, yet economic literature on this phenomenon is limited. For married women with underage children, this living arrangement directly affects the time and energy they can allocate to household care responsibilities, consequently influencing their labor market performance. On the one hand, living with elderly relatives may increase the caregiving burden for married women, negatively impacting their labor market performance in the long run. On the other hand, living with elderly relatives can provide assistance in childcare, thereby alleviating the parenting stress on married women and potentially positively influencing their labor market performance. This living arrangement, living with elderly relatives, will have an impact on women's labor market performance under the combined effects of these two factors.

### 2.1 Grandchild care and female labor market performance

Living with elderly relatives can have a spillover effect by helping married women take care of their children and easing childcare-related stress. The impact of grandchild care on female labor market performance has been empirically tested by numerous scholars using microdata. Ogawa [21] found that in multigenerational families, elderly relatives can take on the role of caring for children, thus promoting female employment. Guo et al. [22] found through empirical research that the Chinese culture of intergenerational support among families helps to enhance not only female labor supply but also family fertility. Based on China Family Panel Studies (CFPS) data, Gu [23] empirically found that living with elderly parents can significantly promote female labor market participation. Using CFPS data, Lu et al. [24] found that the intergenerational care of elderly parents significantly increases their children's labor force participation rate. Using CFPS data from 2010 to 2016, Zou et al. [25] found that grandchild

care significantly increases only the labor participation rate of married women and has no significant effect on the labor participation rate of men. Based on CHFS data from 2011, Sun and Zhou [26] found that living with elderly relatives can effectively relieve the pressure placed on married women to care for their young children, thereby promoting married women's labor market performance. Based on the above literature, this paper proposes the following hypothesis:

Hypothesis H1: Coresident parents can alleviate married women's family care burden through grandchild care, which has a positive impact on women's labor market performance. Thus, coresidence with parents has a spillover effect.

## 2.2 Elderly care and female labor market performance

Living with elderly relatives may increase the level of elderly care-related pressure on married women, resulting in a crowding-out effect. Many scholars have found that family elderly care significantly negatively impacts children's labor market performance. Compared with that of men, family elderly care has a greater negative impact on women's labor market performance. Based on China Health and Nutrition Survey (CHNS) data from 1991 to 2011, Chen et al. [27] found that family elderly care inhibits female labor supply. Becker [28] constructed a time allocation model and found that family care has an impact on women's work but has little effect on that of men. Carmichael and Charles [29] found that caregiving significantly reduces female labor participation and working hours. Lilly et al. [30] found that family care has a negative impact on female labor participation but does not affect female working hours or wage income. Some scholars have found that children who care for their elderly parents have a significant reduction in their income [29, 31]. Based on the above literature, this paper proposes the following hypothesis:

Hypothesis H2: Providing care for coresident parents will lead women to invest much time and energy in family care, which will have a negative impact on their labor market performance. Thus, coresidence with parents has a crowding-out effect.

## 2.3 Overall impact of coresidence with parents on female labor market performance

From the above analysis, it is clear that living with parents has both a crowding-out effect and a spillover effect on female labor supply. Hence, which effect plays the leading role? Based on the current intergenerational relations present in Chinese families, this paper argues that the spillover effect is dominant.

For multi-generational families, there are primarily two forms of intergenerational relationships within the family: one is the support relationship from the younger generation to the older generation, and the other is the nurturing relationship from the older generation to the younger generation. In multi-generational families, these family dynamics often revolve around middle-aged couples, and with the influence of economic development and social policies, intergenerational relationships within the family are also changing. Regarding the relationship between middle-aged couples and their parents, middle-aged couples have grown up, married, and had children, and the nurturing relationship from the older generation to the younger generation no longer exists. As for the support relationship from the younger generation to the older generation, traditional Chinese culture emphasizes "raising children for old-age support," meaning that adult children should take on the responsibility of supporting and caring for their parents in their old age, providing financial assistance and daily care for their elderly parents. However, with economic development and the gradual improvement of social security, the elderly in China no longer primarily require financial support from their children

but rather focus on emotional needs. In multi-generational families, when the elderly are in good health, they may even provide assistance to their own children, including financial support and helping take care of their grandchildren. Regarding the relationship between middle-aged couples and their children, since their children are not yet adults, there is no nurturing relationship from the younger generation to the older generation at this stage. In terms of the nurturing relationship from the older generation to the younger generation, the implementation of the one-child policy has artificially reduced the number of children in each family, elevating the status of children within the family rapidly. The degree of attention that middle-aged couples pay to their young children continues to deepen (while at the same time, the level of attention that the elderly pay to their grandchildren also deepens). In this context, middle-aged couples often devote more energy and financial resources to raising the next generation rather than supporting the previous generation. Therefore, in the context of the new elderly care pattern and family planning policies, on the one hand, middle-aged couples reduce their care for the elderly, and it is more often the elderly who assist their own children, such as helping their children take care of underage grandchildren. On the other hand, middle-aged couples invest more energy and money into raising their own children. This has led to a new pattern of Chinese family relationships known as "preferring the young over the old."

Influenced by traditional culture, women tend to take on a greater share of household caregiving responsibilities. In multigenerational families, the relationship between middle-aged couples and their parents generates "care for the elderly", which increases the caregiving burden on women and results in a crowding-out effect on their employment. As the economy develops and social security gradually improves, the elderly's retirement needs are changing, and they may assist their adult children in taking care of minor children, a phenomenon known as intergenerational caregiving. This can lead to a spillover effect on women's employment. Under the implementation of new elderly care patterns and family planning policies, the traditional concept of "raising children for old-age support" is gradually weakening. It often becomes the case that parents provide support to their children rather than the other way around, reducing the crowding-out effect. In such families, minor grandchildren often hold a high position within the family structure. From the perspective of assisting their children and caring for their grandchildren, elderly individuals may help their middle-aged children by taking care of their grandchildren, thereby reducing women's burden of caring for young children and enhancing the spillover effect. Therefore, this paper proposes the following hypothesis:

Hypothesis H3: The spillover effect of living with parents on female labor market performance is dominant over the crowding-out effect. Therefore, for married families with minor children, living with parents will have a positive impact on the female labor market participation rate and income.

## 3. Research method

### 3.1 Variable source and variable selection

The data in this paper come from the CHFS, which was conducted nationwide by the Southwestern University of Finance and Economics in 2019. The CHFS surveyed 29 provinces, cities, and autonomous regions across China (except Xinjiang, Tibet, Hong Kong, Macao, and Taiwan), which are representative at the national, urban, and rural levels. The data include information on the demographic characteristics, living conditions, income and expenditure, and insurance coverage of households, as well as detailed information on the work and income of family members, providing reliable data support for studying the impact of residential patterns on the labor market performance of married women. The CHFS data are scientifically and randomly sampled, and the survey data are not only representative but also of high quality [32].

In terms of the explained variables, since this paper mainly examines the impact of residential patterns on female labor market performance, labor participation (*work*) and income level (*income*) are selected as the explained variables. If women participate in the labor market, "*work*" takes the value of 1; otherwise, it takes the value of 0. "*Income*" is a continuous variable that represents the total annual after-tax income of married women in the labor market and is logarithmically processed in the paper. In terms of core explanatory variables, this paper selects coresidence (*coresidence*) to measure the family's residential pattern. If a woman is married, has minor children and lives with elderly parents (one or more), then "*coresidence*" takes the value of 1; otherwise, it takes the value of 0. In terms of control variables, this paper draws on the parameters outlined in the literature; the selected control variables include individual-level characteristic variables of married women and family characteristic variables. Among them, the individual characteristic variables include the age (*age*), age squared (*age_square*), ethnicity (*ethnic*), health status (*health*), and years of education (*edu*) of each married women; the family characteristic variables include the husband's income (*income_husband*), the number of minor children (*children_number*), the presence of boys (*boy_number*), and the presence of preschoolers (*preschooler*). In addition, considering that other adults in the family may also have caregiving effects on children, this paper also controls for the number of adults in the family (*adult_number*) except the head couple and the parents of the head couple.

Table 1 defines the variables used in the study, and the summary statistics of the variables used in this study are provided in Table 2. In the process of data processing, to clarify the family relationships, this paper retains only samples with answers provided by the head of household or the head of household's spouse. Furthermore, since this paper discusses the labor performance of married women, the family samples of divorced householders and those of nondivorced families without spouse information are excluded. In addition, since married women's care for their children is more focused during their children's early years, this paper retains only family samples with children under 18 years old. Considering the legal marriage and retirement age of Chinese women, this paper limits the age of married women to 20–55 years old.

As shown in Table 2, approximately of the examined 21.5% of families choose to live with elderly relatives. The average labor force participation rate for women in married families with minor children is 74.5%, and for women participating in the labor market, the logarithmic mean of their income is 10.402. The average age of the women is approximately 40 years old, and most of the women are of Han nationality. Their overall health status is relatively good, and their average education level is high school. Considering the samples of unemployed and employed husbands, the logarithmic mean of husbands' income is 7.075. The average number of children per family is approximately 1.4, with 66.9% of families having boys and 29.1% having preschool children. The average number of adults living together in each family, excluding the head couple and the parents of the head couple, is only 0.03, indicating that only a very small number of families live with others adults.

## 3.2 Model design

**3.2.1 Baseline regression.**  To explore the impact of residential patterns on married women's possibility of participation in the labor market, since labor participation (*work*) is a dummy variable, this paper uses the probit model and OLS model analyses for regression to test the robustness of the results. The paper builds the following models:

$$Pr\left(work_{ij} = 1\right) = \Phi\left(\alpha_0 + \alpha_1 \times coresidence_{ij} + \alpha_2 \times X_{ij} + \eta_j + u_{ij}\right) \quad (1)$$

**Table 1. Variables description.**

| Variable property | Variable name | Description |
|---|---|---|
| Dependent variables | work | If the married woman works, the value is 1; otherwise, it is 0. |
| | income | The logarithm of the total after-tax income for the married woman, including wages, labor remuneration, remuneration for articles, bonuses, cash benefits, subsidies, in-kind income, etc., with deductions for taxes and the five social insurances and housing fund, if any. |
| Independent variables | coresidence | If the woman lives with elderly individuals, the value is 1; otherwise, it is 0. |
| Control variables | age | Age of the married woman. |
| | age_square | The square of the married woman's age divided by 100. |
| | ethnic | If the ethnicity of the married woman is Han, the value is 1; otherwise, it is 0. |
| | health | Health status of the married woman. The variable is measured using the following questions from the questionnaire: How is your current physical condition compared to that of your peers? 1. Very good; 2. Good; 3. Average; 4. Bad; 5. Very bad. |
| | edu | Education level of the married woman. The variable is measured using the following questions from the questionnaire: What is your education level? 1. Have not attended school; 2. Primary school; 3. Junior high school; 4. High school; 5. Special secondary school or vocational high school; 6. Junior college or higher vocational college; 7. Bachelor's degree; 8. Master's degree; 9. Doctoral degree. |
| | income_husband | The logarithm of the husband's total after-tax income. According to the questionnaire setting, if the husband is not employed, he is not required to answer income-related questions. This paper defaults the husband's income to 0 in this type of sample. |
| | children_number | The number of minor children in the family. |
| | boy_number | Whether the family's minor children include boys. If so, the value is 1; otherwise, it is 0. |
| | preschooler | Whether the family contains preschoolers among their minor children. If so, the value is 1; otherwise, it is 0. |
| | adult_number | The number of adults living together in the family, excluding the head couple and the parents of the head couple. |
| Instrumental variable | IV_1 | The proportion of families living with their parents in the county where the family is located (this family is not included in the sample calculation). |
| | IV_2 | Whether the head of household has the concept of "raising children for old-age support." If so, the value is 1; otherwise, it is 0. |

$$work_{ij} = \alpha_0 + \alpha_1 \times coresidence_{ij} + \alpha_2 \times X_{ij} + \eta_j + u_{ij} \qquad (2)$$

The dependent variable $work_{ij}$ refers to the binary dummy variable of whether married women in province $j$ and family $i$ participate in the labor market; $coresidence_{ij}$ is the core explanatory variable of this paper, indicating whether married women live with their parents;

**Table 2. Summary statistics.**

| Variable name | N | Mean | Std. dev. | Min | Max |
|---|---|---|---|---|---|
| work | 6441 | 0.745 | 0.436 | 0.000 | 1.000 |
| income | 3044 | 10.402 | 1.000 | 0.693 | 13.627 |
| coresidence | 6441 | 0.215 | 0.411 | 0.000 | 1.000 |
| age | 6441 | 40.239 | 6.569 | 20.000 | 55.000 |
| age_square | 6441 | 16.623 | 5.278 | 4.000 | 30.250 |
| ethnic | 6441 | 0.880 | 0.326 | 0.000 | 1.000 |
| health | 6441 | 2.448 | 0.908 | 1.000 | 5.000 |
| edu | 6441 | 3.905 | 1.867 | 1.000 | 9.000 |
| income_husband | 6441 | 7.075 | 5.103 | 0.000 | 14.509 |
| children_number | 6441 | 1.424 | 0.593 | 1.000 | 3.000 |
| boy_number | 6441 | 0.669 | 0.471 | 0.000 | 1.000 |
| preschooler | 6441 | 0.291 | 0.454 | 0.000 | 1.000 |
| adult_number | 6441 | 0.030 | 0.196 | 0.000 | 4.000 |
| IV_1 | 6441 | 0.215 | 0.138 | 0.000 | 0.900 |
| IV_2 | 1841 | 0.656 | 0.475 | 0.000 | 1.000 |

$X_{ij}$ is a series of control variables; $\eta_j$ measures province fixed effects; and $u_{ij}$ is the residual term. Due to the obvious development differences between urban and rural areas in China, this paper divides the samples into national, urban, and rural areas for analysis.

After studying the impact of residential patterns on the possibility of female labor market participation, this paper further explores the impact of residential patterns on female employment income. Because only women participating in the labor market have income, there is a problem with sample selection; thus, the Heckman model is used to overcome the sample selection bias. The first-stage equation on the participation of married women in the labor market is as follows:

$$Pr\left(work_{ij} = 1\right) \ = \ \Phi\left(\alpha_0 \ + \ \alpha_1 \ \times \ coresidence_{ij} \ + \ \alpha_2 \ \times \ X_1 \ + \ \eta_j \ + \ u_{ij}\right) \qquad (3)$$

The second-stage of the equation on the impact of residential patterns on the income of married women is as follows:

$$income_{ij} \ = \ \alpha_0 \ + \ \alpha_1 \ \times \ coresidence_{ij} \ + \ \alpha_2 \ \times \ X_2 \ + \ \alpha_3 \ \times \ imr_{ij} \ + \ \eta_j \ + \ u_{ij} \qquad (4)$$

The control variable $X_1$ shown in Eq (3) is consistent with that shown in Eqs (1) and (2). The control variable $X_2$ shown in Eq (4) removes the variable of the husband's wage income (*income_husband*) compared with $X_1$ in Eq (3). The husband's income may affect married women's participation in the labor market; however, for women who have already participated in the labor market, their husband's incomes will not have a direct impact on their income. Therefore, this paper does not include this variable in the second-stage regression. The variable $imr_{ij}$ shown in Eq (4) represents the inverse Mills ratio based on Eq (3), which is placed in Eq (4) as a control variable to correct the sample selection bias.

**3.2.2 Endogenous discussion.** When exploring the impact of residential patterns on women's participation in the labor market, on the one hand, living with parents can alleviate women's parenting-related stress through their parents' grandchild care behavior, thereby affecting women's employment decisions; on the other hand, women who are active in the labor market are more likely to choose to live with their parents to relieve the pressure of childcare, thus creating a two-way causal relationship between residential patterns and women's employment decisions. To alleviate this endogeneity problem, this paper intends to use the instrumental variable method. The instrumental variables need to satisfy the requirement of being highly correlated with the endogenous variables they replace but uncorrelated with the error term, that is, to find the variables related to whether they live with their parents and uncorrelated with whether women participate in the labor market as instrumental variables. Based on this, this paper selects two variables as instrumental variables. The first instrumental variable (*IV_1*) is the proportion of families living with parents in the county where the family is located (excluding the family itself when calculating the corresponding proportion of each family). On the one hand, the decision to live with parents is influenced by many factors, including the deeply rooted Chinese sentiment of "raising children for old-age support"; this concept has distinct regional characteristics. Therefore, in areas with a heavy cultural concept of "raising children for old-age support," the probability of adult children living with their parents may be higher. The proportion of families living with their parents in each district and county is related to whether a single family lives with their parents. On the other hand, the proportion of families living with parents is not related to the employment decisions of married women in such families; thus, the variable satisfies the relevant assumptions of the instrument. The use of regional-level indicators as instrumental variables for individual-level indicators is a common method of selecting instrumental variables in the literature [33].

In addition, this paper constructs the second instrumental variable (*IV_2*) based on a direct question in the questionnaire that measures whether the household head holds the traditional concept of "raising children for old-age support." If the household head holds this traditional concept, the variable takes a value of 1; otherwise, it takes a value of 0. Since the household head is the decision-maker for family affairs, their beliefs will influence the decision of whether the family lives with parents, so we examined whether the household head holds the traditional concept of "raising children for old-age support" and used it as an instrumental variable. If the household head holds this concept, the family is more likely to live with the elderly for caretaking and support, but the household head's belief in "raising children for old-age support" is not directly related to the employment decisions of family females. Therefore, this variable meets the requirements of being directly related to the endogenous variable and uncorrelated with the disturbance term. The 2015 CHFS questionnaire measured whether the household head holds the traditional concept of "raising children for old-age support" with the following question: "What do you think is the main reason for raising children? 1. To carry on the family line; 2. Affection for children based on emotional considerations; 3. To support the elderly with children; 4. To maintain marital stability; 5. Other." If the household head's choice includes the third option, then it is considered that the household head holds the traditional concept of "raising children for old-age support." The CHFS conducts surveys every two years, including some tracking samples and new samples, with questionnaire settings differing from previous years. Since the 2019 questionnaire did not include questions related to "raising children for old-age support," and the 2015 questionnaire did not directly measure whether family members live together, we matched the data from 2015 with the data from 2019 using the unique ID of each family, obtaining a total of 1,841 data points for further endogeneity analysis.

When exploring the impact of residential patterns on women's employment income, there is a sample selection problem because only women who participate in the labor market have income; thus, this paper uses the Heckman two-stage model to mitigate the endogeneity problem caused by sample selection bias. Husbands' income may affect whether married women participate in the labor market; however, for women who have already participated in the labor market, their husband's income will not have a direct impact on their income. Thus, the variable representing the husband's income (*income_husband*) meets the requirements of relevance and exclusivity and is thus chosen as an exclusionary variable in this paper. The first-stage regression is a probit model that includes the full sample, which is used to estimate the probability of married women participating in the labor market and calculate the inverse Mills ratio for each sample. The second-stage regression includes only the sample participating in the labor market and uses the inverse Mills ratio obtained in the first stage as a control variable to correct for sample selection bias.

## 4. Empirical analysis

### 4.1 Baseline regression

**4.1.1 The impact of residential patterns on female labor market participation.** Table 3 shows the regression results of the impact of residential patterns on the possibility of female labor market participation. Specifically, taking the regression results of the probit model as an example, for the national and urban samples, the labor force participation rate of women in families with parents increases by 5.3% and 5.6%, respectively, compared with that of women in families without parents. For women in rural families, whether they live with their parents does not have a significant effect on their labor force participation rate. As seen from Table 3, coresidence with parents significantly increases the possibility of female labor market participation in both national and urban samples; thus, Hypothesis H3 is verified. However, in the

**Table 3. The impact of residential patterns on female labor market participation.**

|  | The national samples | | The urban samples | | The rural samples | |
|---|---|---|---|---|---|---|
|  | **Probit** | **OLS** | **Probit** | **OLS** | **Probit** | **OLS** |
| coresidence | 0.053*** | 0.052*** | 0.056*** | 0.057*** | 0.019 | 0.015 |
|  | (3.967) | (4.107) | (3.290) | (3.497) | (0.919) | (0.742) |
| age | 0.041*** | 0.046*** | 0.051*** | 0.056*** | 0.029** | 0.038** |
|  | (4.778) | (4.716) | (4.893) | (4.708) | (2.021) | (2.275) |
| age_square | -0.047*** | -0.052*** | -0.061*** | -0.067*** | -0.029* | -0.039** |
|  | (-4.294) | (-4.277) | (-4.666) | (-4.480) | (-1.654) | (-1.986) |
| ethnic | -0.023 | -0.025 | 0.025 | 0.024 | -0.046* | -0.048* |
|  | (-1.257) | (-1.356) | (1.022) | (0.947) | (-1.671) | (-1.826) |
| health | -0.021*** | -0.022*** | -0.013* | -0.015* | -0.046*** | -0.046*** |
|  | (-3.450) | (-3.558) | (-1.710) | (-1.890) | (-4.625) | (-4.504) |
| edu | 0.041*** | 0.041*** | 0.054*** | 0.053*** | 0.028*** | 0.030*** |
|  | (11.906) | (12.327) | (13.758) | (13.642) | (2.816) | (3.035) |
| income_husband | -0.006*** | -0.006*** | -0.005*** | -0.005*** | -0.007*** | -0.007*** |
|  | (-5.528) | (-5.810) | (-3.469) | (-3.730) | (-3.610) | (-3.556) |
| children_number | -0.025** | -0.027** | -0.041*** | -0.045*** | -0.013 | -0.014 |
|  | (-2.387) | (-2.404) | (-3.054) | (-3.041) | (-0.839) | (-0.849) |
| boy_number | 0.019* | 0.020* | 0.026* | 0.027** | -0.010 | -0.009 |
|  | (1.673) | (1.683) | (1.922) | (1.962) | (-0.468) | (-0.415) |
| preschooler | -0.115*** | -0.118*** | -0.115*** | -0.113*** | -0.114*** | -0.127*** |
|  | (-7.822) | (-7.429) | (-6.571) | (-6.158) | (-4.380) | (-3.962) |
| adult_number | 0.013 | 0.012 | 0.059 | 0.057 | -0.035 | -0.042 |
|  | (0.493) | (0.442) | (1.441) | (1.548) | (-1.098) | (-1.140) |
| _cons |  | -0.204 |  | -0.477** |  | -0.286 |
|  |  | (-1.042) |  | (-2.044) |  | (-0.668) |
| Observations | 6441 | 6441 | 4616 | 4616 | 1825 | 1825 |
| Province FE | Yes | Yes | Yes | Yes | Yes | Yes |
| Adj-R2 |  | 0.070 |  | 0.086 |  | 0.096 |

Note: (1) *, **, and *** represent significance at the 10%, 5%, and 1% levels, respectively. (2) The numbers shown in parentheses for the probit model are z values under robust standard errors, and the numbers shown in parentheses for the OLS model are t values under robust standard errors. (3) The probit model results report the average marginal effects.

rural samples, this effect is no longer significant. The reason may be that urban residents often live separately from their parents after they get married. Therefore, for urban families with minor children, grandchild care has become one of the most important reasons for living with parents. For urban families, the spillover effect of living with parents is more significant than the crowding-out effect, which can significantly increase the possibility of married women's labor market participation. The security system of urban families is not as well developed as that for urban families; thus, rural families place more emphasis on the extended family system. Under the extended family system, members share risks and benefits, similar to an insurance mechanism. Therefore, people in rural families still tend to live with their parents after marriage, and this residential decision is not related to the presence or absence of children or whether children are young. In addition, women in rural families are more engaged in agricultural work, which means that the residential pattern of coresidence with parents brings less of a spillover effect to women. Therefore, living with parents does not significantly impact rural women's labor market participation.

As far as the regression results of control variables are concerned, taking the regression results of the probit model as an example, in the national sample, there is an inverted U-shaped relationship between the age of married women and the possibility of their participation in the labor market; this indicates that the possibility of their participation in the labor market increases and then decreases as their age increases. Thus, the better the health status of a woman is, the higher the possibility of her participating in the labor market is; in addition, the higher the education level of a woman is, the higher the possibility of her participating in the labor market is. Furthermore, her husband's income will have a significant negative impact on a woman's labor market participation, and for each additional child, the likelihood of a woman participating in the labor market decreases by 2.7 percent. According to the regression results, if her family's minor children include boys, the possibility of a woman's participation in the labor market increases by 2%. This may be because the pattern of there being more men than women in the marriageable population since the 1980s has caused a squeeze in the marriage market, making it more difficult for men to be matched in. In the face of high marriage expenses, a man's family must make material preparations in advance; thus, the presence of a boy in the family promotes the active participation of women in the labor market. The regression results show that women are 11.8% less likely to participate in the labor market if their minor children include preschoolers compared with families without preschoolers. This finding may be due to the heavier burden of care for preschool children, which may hinder women's participation in the labor market. In addition, the number of adults living in the family, excluding the head couple and the parents of the head couple, does not have a significant impact on women's labor market participation. According to the results of statistics, there are only 0.03 other adults in each household, and only 2.6% of the samples have other adults living with their families. Thus, the number of other coresiding adults does not have an impact on women's labor market participation.

The regression results of the control variables for the urban sample are similar to those of the national sample and thus will not be repeated here. The regression results for the control variables in the rural sample are basically similar to those of the national and urban samples; however, the likelihood of participation in the labor market in the rural sample decreases by 4.6% (4.8%) if the woman is of Han ethnicity. This may be because the proportion of ethnic minorities in rural areas is higher than that in urban areas. Although ethnic minority women in rural areas are more likely to engage in agricultural labor, their labor force participation rate is higher. Therefore, if a woman in the rural sample is Han, her ethnicity will have a negative impact on her labor market participation. In addition, the number of children (*children_number*) and whether there are boys in the family (*boy_number*) are no longer significant in the rural sample. This may be because rural families do not take care of their children as meticulously as urban families; thus, the number of children does not have a significant impact on women's participation in the labor market. Rural families believe more in the concept of "raising children for old-age support" and hope that their children can support their families when they become adults. In addition, in contrast to urban families, rural families do not prepare in advance for their sons to enter the marriage market, such as by providing them with houses and cars to improve their competitiveness. Therefore, whether there are boys in the family does not have a significant impact on the participation of rural women in the labor market.

In addition, the regression results show that the results obtained from the probit and OLS models are very close, indicating that the conclusions are robust.

**4.1.2 The impact of residential patterns on female labor market income.** Table 4 shows the impact of residential patterns on women's labor market earnings using the Heckman two-stage model. The first-stage regression results for national and urban samples indicate that living with parents significantly promotes women's labor market participation. The inverse Mills

**Table 4. The impact of residential patterns on female labor market income.**

| | The national samples | | The urban samples | | The rural samples | |
|---|---|---|---|---|---|---|
| | **first** | **second** | **first** | **second** | **first** | **second** |
| coresidence | 0.176*** | 0.089** | 0.185*** | 0.167*** | 0.073 | -0.060 |
| | (3.954) | (1.966) | (3.278) | (3.297) | (0.920) | (-0.621) |
| age | 0.139*** | 0.161*** | 0.169*** | 0.212*** | 0.114** | 0.060 |
| | (4.757) | (4.330) | (4.860) | (4.780) | (2.011) | (0.617) |
| age_square | -0.156*** | -0.190*** | -0.202*** | -0.256*** | -0.114* | -0.083 |
| | (-4.279) | (-4.193) | (-4.639) | (-4.677) | (-1.648) | (-0.727) |
| ethnic | -0.078 | -0.045 | 0.081 | 0.016 | -0.182* | 0.041 |
| | (-1.256) | (-0.827) | (1.022) | (0.281) | (-1.664) | (0.292) |
| health | -0.071*** | -0.139*** | -0.043* | -0.128*** | -0.182*** | -0.135** |
| | (-3.439) | (-6.567) | (-1.708) | (-5.875) | (-4.549) | (-2.005) |
| edu | 0.138*** | 0.341*** | 0.177*** | 0.391*** | 0.108*** | 0.153*** |
| | (11.561) | (17.046) | (13.106) | (13.846) | (2.794) | (3.365) |
| income_husband | -0.020*** | | -0.015*** | | -0.027*** | |
| | (-5.489) | | (-3.453) | | (-3.583) | |
| children_number | -0.084** | -0.113*** | -0.136*** | -0.143*** | -0.053 | -0.114 |
| | (-2.384) | (-3.086) | (-3.046) | (-3.127) | (-0.838) | (-1.529) |
| boy_number | 0.065* | 0.017 | 0.086* | 0.037 | -0.038 | 0.088 |
| | (1.673) | (0.503) | (1.920) | (1.006) | (-0.468) | (0.924) |
| preschooler | -0.384*** | -0.301*** | -0.378*** | -0.366*** | -0.447*** | -0.122 |
| | (-7.729) | (-4.408) | (-6.491) | (-5.157) | (-4.336) | (-0.688) |
| adult_number | 0.043 | 0.087 | 0.193 | 0.241*** | -0.139 | -0.086 |
| | (0.493) | (1.008) | (1.440) | (2.880) | (-1.097) | (-0.383) |
| imr | | 1.536*** | | 1.968*** | | 0.078 |
| | | (4.670) | | (5.340) | | (0.113) |
| _cons | -2.140*** | 6.108*** | -2.938*** | 4.742*** | -2.198* | 9.614*** |
| | (-3.642) | (7.286) | (-4.277) | (4.619) | (-1.665) | (4.002) |
| Observations | 6441 | 3044 | 4616 | 2534 | 1825 | 510 |
| Province FE | Yes | Yes | Yes | Yes | Yes | Yes |
| Pseudo R2 | 0.068 | | 0.082 | | 0.114 | |
| Adj-R2 | | 0.362 | | 0.338 | | 0.114 |

Note: (1) *, **, and *** represent significance at the 10%, 5%, and 1% levels, respectively. (2) The numbers shown in parentheses in the first stage are z values under robust standard errors, and the numbers shown in parentheses in the second stage are t values under robust standard errors. (3) The first stage shown in the table reports the regression coefficients of the probit model and not the average marginal effects.

ratios are significant at the 1% level , indicating that the samples have the problem of selection bias. In the second stage, the regression results for the national and urban samples show that living with parents increases women's employment income by 8.9% and 16.7%, respectively, compared to that found for women with families who do not live with their parents; thus, Hypothesis H3 is supported. For the rural samples, the core explanatory variables in both the first and second stages are nonsignificant. The inverse Mills ratio is also nonsignificant, indicating that the examined residential pattern has no significant impact on rural women's labor market participation or income and that there is no sample selection bias.

For urban families, the spillover effect of living with parents is more significant than the crowding-out effect, which not only significantly promotes women's labor market participation but also enables women to spend more energy on work and thus obtain higher income

returns. For rural families, women are more engaged in agricultural activities, which means that the examined residential pattern has no significant impact on the income of women in rural families.

## 4.2 Mechanism analysis

Based on the literature, this paper argues that for women who are married and have minor children, there is a spillover effect and a crowding-out effect of coresidence with their parents. The combined effect of these two effects impacts women's labor market participation and labor market income. Next, this paper will further test these two effects.

**4.2.1 Test of the spillover effect.** When living with elderly parents, these elderly parents can care for minor children. Grandchild care will help relieve women's caregiving-related pressure, thereby promoting women's labor market participation and income. Based on this, this paper investigates the spillover effect further by regressing a sample consisting of women who are married without children and those who are married with children who are adults.

According to the regression results shown in Table 5, the core explanatory variable is no longer significant in the sample of women without children and those who are married with adult children, indicating that if a family has no children or the family's children are already adults, the factor of grandchild care will no longer be significant when living with parents. At this time, the residential pattern of coresidence with parents no longer promotes female labor market participation. The regression results show that the promotion effect of coresidence with parents on female labor market participation is indeed due to the factor of grandchild care; thus, Hypothesis H1 is verified.

According to the regression results shown in Table 6, after correcting the sample selection bias, the coresidence model no longer significantly affects women's income for the sample of women without children and those who are married with adult children. The empirical results in this section show that the coresidence model promotes women's labor market earnings mainly because the elderly relatives share part of the childcare-related stress in families with minor children; thus, Hypothesis H1 is verified.

**4.2.2 Test of the crowding-out effect.** In the pattern of coresidence with parents, in addition to the spillover effect of grandchild care, women may also spend more time and energy caring for aging parents, which has a crowding-out effect on women's labor market participation and income. Due to limitations of the utilized questionnaire, this paper cannot obtain specific information related to the situation of parents when they are not included in a larger family; the questionnaire did not directly query related issues of elderly care. Therefore, this paper selects the average health status of parents (*coresidence_health*) from the sample of those who engage in coresidence with parents as an indicator for measuring elderly care and testing the crowding-out effect. According to the questionnaire, the higher the value of the average health status is, the worse the parents' health status is, and the heavier the burden of elderly care for married women is.

According to the regression results shown in Table 7, the results are nonsignificant and negative for both the national and rural samples. However, for the urban sample, the results are significantly negative, indicating that the heavier the burden of elderly care is, the less likely women are to participate in the labor market; thus, Hypothesis H2 is verified. The regression results shown in Table 8 are similar to those shown in Table 7 and will thus not be repeated here.

According to the abovementioned results, it can be seen that the spillover effect and the crowding-out effect coexist when married couples with children live with parents; however, since the spillover effect is greater than the crowding out effect, the coresidence model has a

**Table 5. The impact of residential patterns on female labor market participation (sample of women with no children and those who are married with adult children).**

| | The national samples | | The urban samples | | The rural samples | |
|---|---|---|---|---|---|---|
| | **Probit** | **OLS** | **Probit** | **OLS** | **Probit** | **OLS** |
| coresidence | 0.025 | 0.021 | 0.021 | 0.019 | 0.004 | 0.000 |
| | (1.624) | (1.405) | (0.940) | (0.865) | (0.218) | (0.007) |
| age | 0.065*** | 0.066*** | 0.090*** | 0.093*** | 0.062*** | 0.072*** |
| | (7.021) | (7.148) | (8.188) | (8.469) | (3.205) | (3.008) |
| age_square | -0.085*** | -0.086*** | -0.116*** | -0.120*** | -0.072*** | -0.082*** |
| | (-8.116) | (-8.190) | (-9.258) | (-9.429) | (-3.479) | (-3.248) |
| ethnic | -0.049** | -0.045** | -0.029 | -0.028 | -0.028 | -0.026 |
| | (-2.222) | (-2.297) | (-0.947) | (-0.967) | (-1.006) | (-1.064) |
| health | -0.032*** | -0.033*** | -0.032*** | -0.034*** | -0.051*** | -0.050*** |
| | (-5.537) | (-5.426) | (-4.009) | (-4.082) | (-6.428) | (-6.157) |
| edu | 0.022*** | 0.021*** | 0.052*** | 0.048*** | 0.004 | 0.005 |
| | (5.365) | (5.400) | (10.528) | (10.447) | (0.419) | (0.523) |
| income_husband | -0.007*** | -0.007*** | -0.004*** | -0.004*** | -0.004*** | -0.004*** |
| | (-6.020) | (-5.993) | (-2.809) | (-2.804) | (-2.764) | (-2.697) |
| children_number | 0.008 | 0.010 | 0.013 | 0.019 | 0.007 | 0.008 |
| | (0.732) | (0.905) | (0.876) | (1.227) | (0.490) | (0.534) |
| boy_number | 0.061*** | 0.059*** | 0.064*** | 0.063*** | 0.033 | 0.029 |
| | (4.130) | (4.074) | (3.410) | (3.358) | (1.544) | (1.401) |
| preschooler | - | - | - | - | - | - |
| adult_number | -0.050*** | -0.050*** | -0.052** | -0.062** | -0.053** | -0.049** |
| | (-2.928) | (-2.683) | (-2.057) | (-2.157) | (-2.551) | (-2.107) |
| _cons | | -0.339* | | -0.991*** | | -0.593 |
| | | (-1.650) | | (-4.217) | | (-1.034) |
| Observations | 6644 | 6644 | 4090 | 4090 | 2554 | 2554 |
| Province FE | Yes | Yes | Yes | Yes | Yes | Yes |
| Adj-R2 | | 0.064 | | 0.112 | | 0.063 |

Note: (1) *, **, and *** represent significance at the 10%, 5%, and 1% levels, respectively. (2) The numbers shown in parentheses for the probit model are z values under robust standard errors, and the numbers shown in parentheses for the OLS model are t values under robust standard errors. (3) The probit model results report the average marginal effects.

significant effect on the possibility of women's participation in the labor market and income in the national and urban samples.

## 5. Further analysis

### 5.1 Endogeneity problem

When discussing the impact of residential patterns on the possibility of female labor market participation, there may be a two-way causal relationship between residential patterns and female employment decisions, resulting in an endogeneity problem. To alleviate this endogeneity problem, this paper uses the instrumental variable method. This paper selects the proportion of families living with parents in the county where the family is located ($IV\_1$) and whether the household head holds the traditional concept of "raising children for old-age support" ($IV\_2$) as the instrumental variables. Generally, the two-stage least squares method and IV-probit can be used to estimate instrumental variables. However, in the model used in this paper, residential patterns and labor participation are discrete variables; thus, the two-stage

**Table 6. The impact of residential patterns on female labor market income (sample of women with no children and those who are married with adult children).**

| | The national samples | | The urban samples | | The rural samples | |
|---|---|---|---|---|---|---|
| | first | second | first | second | first | second |
| coresidence | 0.080 | 0.001 | 0.062 | -0.014 | 0.017 | 0.070 |
| | (1.624) | (0.010) | (0.940) | (-0.242) | (0.218) | (0.786) |
| age | 0.204*** | 0.116*** | 0.274*** | 0.092* | 0.245*** | -0.053 |
| | (6.949) | (2.823) | (8.007) | (1.677) | (3.182) | (-0.278) |
| age_square | -0.266*** | -0.148*** | -0.355*** | -0.119* | -0.283*** | 0.058 |
| | (-8.006) | (-2.858) | (-8.996) | (-1.690) | (-3.451) | (0.283) |
| ethnic | -0.153** | 0.092 | -0.089 | 0.049 | -0.109 | 0.204 |
| | (-2.220) | (1.118) | (-0.946) | (0.524) | (-1.005) | (1.234) |
| health | -0.102*** | -0.102*** | -0.097*** | -0.059** | -0.200*** | 0.018 |
| | (-5.503) | (-4.092) | (-3.985) | (-2.172) | (-6.334) | (0.191) |
| edu | 0.069*** | 0.304*** | 0.157*** | 0.308*** | 0.016 | 0.144*** |
| | (5.343) | (21.614) | (10.251) | (10.755) | (0.419) | (3.181) |
| income_husband | -0.021*** | | -0.013*** | | -0.017*** | |
| | (-5.976) | | (-2.801) | | (-2.759) | |
| children_number | 0.025 | 0.019 | 0.039 | 0.051 | 0.027 | -0.050 |
| | (0.731) | (0.553) | (0.876) | (1.234) | (0.490) | (-0.769) |
| boy_number | 0.191*** | 0.083* | 0.195*** | 0.042 | 0.129 | -0.011 |
| | (4.117) | (1.716) | (3.396) | (0.810) | (1.544) | (-0.100) |
| preschooler | - | - | - | - | - | - |
| adult_number | -0.158*** | -0.069 | -0.159** | -0.096 | -0.207** | 0.249 |
| | (-2.923) | (-0.793) | (-2.053) | (-0.943) | (-2.551) | (1.441) |
| imr | | 0.814** | | 0.679* | | -1.908* |
| | | (2.313) | | (1.752) | | (-1.900) |
| _cons | -2.646*** | 7.318*** | -4.463*** | 7.769*** | -3.716** | 11.820*** |
| | (-4.025) | (8.172) | (-5.986) | (6.372) | (-2.008) | (2.627) |
| Observations | 6644 | 2458 | 4090 | 1877 | 2554 | 581 |
| Province FE | Yes | Yes | Yes | Yes | Yes | Yes |
| Pseudo R2 | 0.061 | | 0.100 | | 0.080 | |
| Adj-R2 | | 0.364 | | 0.356 | | 0.196 |

Note: (1) *, **, and *** represent significance at the 10%, 5%, and 1% levels, respectively. (2) The numbers shown in parentheses in the first stage are z values under robust standard errors, and the numbers shown in parentheses in the second stage are t values under robust standard errors. (3) The first stage shown in the table reports the regression coefficients of the probit model and not the average marginal effects.

least squares method and the IV-probit estimation method may fail. Thus, this paper adopts the conditional mixed process (CMP) method proposed by Roodman [14] to re-estimate the model. The CMP is suitable for estimation using instrumental variables when both the independent and dependent variables are discrete variables; this method has been recognized and widely used in academia [34].

This paper uses the CMP method for two-stage regression, and Table 9 shows the two-stage regression results using *IV_1*. According to Table 9, the instrumental variable (*IV_1*) is significantly positively correlated with whether a family lives with parents at the 1% level for the national and urban samples, while the rural samples are significantly positively correlated at the 5% level; this indicates that the variable satisfies the correlation condition for an instrumental variable. Considering that families may migrate to counties with a high proportion of families living with parents, this will affect the correlation of the instrumental variables. In this

**Table 7. The impact of elderly care on female labor market participation.**

| | The national samples | | The urban samples | | The rural samples | |
|---|---|---|---|---|---|---|
| | **Probit** | **OLS** | **Probit** | **OLS** | **Probit** | **OLS** |
| coresidence_health | -0.018 | -0.017 | -0.040** | -0.042** | -0.005 | -0.004 |
| | (-1.371) | (-1.237) | (-2.201) | (-2.067) | (-0.266) | (-0.230) |
| age | 0.031* | 0.038* | 0.034 | 0.041 | 0.024 | 0.029 |
| | (1.846) | (1.947) | (1.488) | (1.490) | (0.940) | (1.003) |
| age_square | -0.032 | -0.040 | -0.035 | -0.043 | -0.024 | -0.030 |
| | (-1.467) | (-1.635) | (-1.176) | (-1.233) | (-0.740) | (-0.859) |
| ethnic | -0.071** | -0.073** | 0.040 | 0.048 | -0.146*** | -0.128*** |
| | (-2.010) | (-2.067) | (0.742) | (0.792) | (-2.748) | (-2.888) |
| health | -0.010 | -0.011 | 0.025 | 0.028 | -0.054*** | -0.055*** |
| | (-0.698) | (-0.724) | (1.160) | (1.142) | (-2.656) | (-2.638) |
| edu | 0.036*** | 0.037*** | 0.045*** | 0.044*** | 0.043*** | 0.042*** |
| | (4.519) | (4.704) | (4.685) | (4.533) | (2.678) | (2.622) |
| income_husband | -0.003 | -0.003 | 0.001 | 0.001 | -0.008** | -0.008** |
| | (-1.277) | (-1.441) | (0.430) | (0.224) | (-2.464) | (-2.369) |
| children_number | -0.004 | -0.007 | -0.027 | -0.032 | 0.007 | 0.004 |
| | (-0.206) | (-0.303) | (-0.941) | (-1.004) | (0.230) | (0.114) |
| boy_number | -0.028 | -0.027 | -0.004 | -0.004 | -0.055 | -0.048 |
| | (-1.117) | (-1.106) | (-0.121) | (-0.136) | (-1.377) | (-1.243) |
| preschooler | -0.080*** | -0.082*** | -0.039 | -0.040 | -0.171*** | -0.170*** |
| | (-2.732) | (-2.594) | (-0.960) | (-0.953) | (-3.909) | (-3.351) |
| adult_number | -0.003 | -0.005 | 0.055 | 0.066 | 0.002 | -0.019 |
| | (-0.093) | (-0.128) | (0.885) | (1.033) | (0.038) | (-0.413) |
| _cons | | 0.097 | | -0.180 | | 0.531 |
| | | (0.253) | | (-0.350) | | (0.917) |
| Observations | 1384 | 1384 | 817 | 817 | 534 | 567 |
| Province FE | Yes | Yes | Yes | Yes | Yes | Yes |
| Adj-R2 | | 0.071 | | 0.070 | | 0.130 |

Note: (1) *, **, and *** represent significance at the 10%, 5%, and 1% levels, respectively. (2) The numbers shown in parentheses for the probit model are z values under robust standard errors, and the numbers shown in parentheses for the OLS model are t values under robust standard errors. (3) The probit model results report the average marginal effects.

paper, the instrumental variable regressions are constructed again after excluding the sample of families with migratory behavior, and the instrumental variable is still highly correlated with the core explanatory variables. The results of the regression using the instrumental variable after removing the sample of families with migratory behavior are placed in S1 Appendix. In the national and urban samples, the endogeneity test parameter *atanhrho_12* is significant at the 1% and 5% level, respectively. The results also show that the core explanatory variable is endogenous and that the CMP method produces results that are better than those estimated directly using the probit or OLS model. In the rural samples, *atanhrho_12* is nonsignificant, indicating that the core explanatory variables in the rural samples are not significantly endogenous. The estimation results of the probit and OLS models can be directly referred to. After correcting the possible endogenous bias in the national and urban samples, there is still a significant positive relationship between coresidence with parents and women's labor market participation, which is significant at the 1% level. Specifically, compared with families who do

**Table 8. The impact of elderly care on female labor market income.**

| | The national samples | | The urban samples | | The rural samples | |
|---|---|---|---|---|---|---|
| | **first** | **second** | **first** | **second** | **first** | **second** |
| coresidence_health | -0.065 | -0.106[*] | -0.146[**] | -0.063 | -0.021 | -0.047 |
| | (-1.370) | (-1.916) | (-2.180) | (-0.759) | (-0.266) | (-0.488) |
| age | 0.116[*] | 0.213[**] | 0.126 | 0.133 | 0.098 | 0.201 |
| | (1.838) | (2.397) | (1.482) | (1.288) | (0.936) | (1.043) |
| age_square | -0.118 | -0.241[**] | -0.129 | -0.173 | -0.097 | -0.231 |
| | (-1.463) | (-2.190) | (-1.172) | (-1.372) | (-0.737) | (-0.952) |
| ethnic | -0.263[**] | -0.076 | 0.145 | 0.223 | -0.597[***] | -0.376 |
| | (-2.002) | (-0.489) | (0.741) | (1.296) | (-2.708) | (-1.084) |
| health | -0.038 | -0.105[*] | 0.093 | -0.011 | -0.219[***] | -0.235 |
| | (-0.698) | (-1.849) | (1.155) | (-0.151) | (-2.608) | (-1.474) |
| edu | 0.133[***] | 0.413[***] | 0.164[***] | 0.299[***] | 0.177[***] | 0.369[***] |
| | (4.408) | (8.827) | (4.538) | (4.120) | (2.636) | (3.957) |
| income_husband | -0.011 | | 0.005 | | -0.034[**] | |
| | (-1.274) | | (0.430) | | (-2.432) | |
| children_number | -0.015 | -0.069 | -0.100 | -0.039 | 0.028 | -0.107 |
| | (-0.206) | (-0.994) | (-0.940) | (-0.451) | (0.230) | (-0.674) |
| boy_number | -0.103 | -0.082 | -0.014 | 0.086 | -0.226 | -0.412[**] |
| | (-1.115) | (-0.997) | (-0.121) | (0.975) | (-1.374) | (-2.008) |
| preschooler | -0.295[***] | -0.436[***] | -0.141 | -0.138 | -0.699[***] | -0.562 |
| | (-2.710) | (-3.185) | (-0.957) | (-1.181) | (-3.828) | (-1.367) |
| adult_number | -0.012 | -0.104 | 0.200 | 0.051 | 0.006 | -0.017 |
| | (-0.093) | (-0.765) | (0.882) | (0.278) | (0.038) | (-0.062) |
| imr | | 2.992[***] | | 0.805 | | 2.041[*] |
| | | (3.568) | | (0.762) | | (1.706) |
| _cons | -0.925 | 4.796[***] | -1.954 | 6.903[***] | 0.326 | 5.528 |
| | (-0.713) | (2.597) | (-1.196) | (2.871) | (0.159) | (1.491) |
| Observations | 1384 | 610 | 817 | 462 | 534 | 140 |
| Province FE | Yes | Yes | Yes | Yes | Yes | Yes |
| Pseudo R2 | 0.094 | | 0.109 | | 0.177 | |
| Adj-R2 | | 0.386 | | 0.362 | | 0.071 |

Note: (1) *, **, and *** represent significance at the 10%, 5%, and 1% levels, respectively. (2) The numbers shown in parentheses in the first stage are z values under robust standard errors, and the numbers shown in parentheses in the second stage are t values under robust standard errors. (3) The first stage shown in the table reports the regression coefficients of the probit model and not the average marginal effects.

not coresidence with parents, those who do coresidence with parents experience an increase in the possibility of women's labor market participation of 31.6% and 31.4%, respectively.

The regression results shown in Table 10 are similar to those shown in Table 9. Because the sample size is significantly reduced after matching the 2015 data with the 2019 data, the results no longer converge after the subsample regression, so this paper only presents the regression results for the national sample. According to the regression results, the heavier the concept of "raising children for old-age support" of the household head, the more likely the family will live with parents, and it is significant at the 5% level, indicating that this variable meets the correlation conditions. The endogeneity test parameter atanhrho_12 is significant at the 1% level, indicating that the core explanatory variable is endogenous. After correcting for possible

**Table 9. The CMP estimation results of the impact of residential patterns on female labor market participation (*IV_1*).**

| | The national samples | | | The urban samples | | | The rural samples | | |
|---|---|---|---|---|---|---|---|---|---|
| | first | second | margin | first | second | margin | first | second | margin |
| IV_1 | 1.194*** | | | 1.178*** | | | 0.468* | | |
| | (8.619) | | | (6.480) | | | (1.952) | | |
| coresidence | | 1.052*** | 0.316*** | | 1.030*** | 0.314*** | | 0.564 | 0.145 |
| | | (4.903) | (4.711) | | (3.360) | (3.269) | | (1.334) | (1.296) |
| age | 0.054* | 0.116*** | 0.035*** | 0.046 | 0.151*** | 0.046*** | 0.076 | 0.102* | 0.026* |
| | (1.764) | (3.842) | (3.896) | (1.181) | (4.130) | (4.204) | (1.366) | (1.757) | (1.782) |
| age_square | -0.095** | -0.121*** | -0.036*** | -0.082* | -0.175*** | -0.053*** | -0.138** | -0.093 | -0.024 |
| | (-2.464) | (-3.176) | (-3.215) | (-1.682) | (-3.745) | (-3.810) | (-2.011) | (-1.276) | (-1.291) |
| ethnic | -0.082 | -0.035 | -0.011 | -0.123 | 0.110 | 0.034 | 0.008 | -0.173 | -0.044 |
| | (-1.376) | (-0.575) | (-0.576) | (-1.488) | (1.390) | (1.387) | (0.090) | (-1.605) | (-1.615) |
| health | 0.040* | -0.078*** | -0.023*** | 0.054** | -0.053** | -0.016** | 0.012 | -0.180*** | -0.046*** |
| | (1.937) | (-3.932) | (-3.944) | (2.002) | (-2.160) | (-2.158) | (0.352) | (-4.512) | (-4.644) |
| edu | -0.028** | 0.137*** | 0.041*** | -0.009 | 0.169*** | 0.051*** | 0.118*** | 0.085* | 0.022** |
| | (-2.277) | (11.394) | (12.020) | (-0.622) | (11.084) | (12.072) | (3.404) | (1.918) | (1.968) |
| income_husband | -0.012*** | -0.015*** | -0.005*** | -0.009** | -0.012** | -0.004** | -0.013* | -0.024*** | -0.006*** |
| | (-3.322) | (-3.691) | (-3.762) | (-2.076) | (-2.499) | (-2.528) | (-1.894) | (-2.994) | (-3.087) |
| children_number | 0.166*** | -0.126*** | -0.038*** | 0.178*** | -0.174*** | -0.053*** | 0.060 | -0.061 | -0.016 |
| | (4.684) | (-3.619) | (-3.598) | (3.787) | (-3.909) | (-3.894) | (1.050) | (-0.978) | (-0.978) |
| boy_number | -0.016 | 0.064* | 0.019* | -0.000 | 0.079* | 0.024* | -0.071 | -0.025 | -0.006 |
| | (-0.411) | (1.719) | (1.722) | (-0.004) | (1.818) | (1.823) | (-0.945) | (-0.306) | (-0.307) |
| preschooler | 0.035 | -0.356*** | -0.107*** | 0.034 | -0.356*** | -0.108*** | 0.091 | -0.450*** | -0.116*** |
| | (0.696) | (-7.035) | (-7.249) | (0.561) | (-5.989) | (-6.156) | (0.937) | (-4.415) | (-4.467) |
| adult_number | 0.554*** | -0.136 | -0.041 | 0.594*** | -0.005 | -0.002 | 0.452*** | -0.211 | -0.054 |
| | (5.850) | (-1.395) | (-1.387) | (4.068) | (-0.034) | (-0.034) | (3.631) | (-1.590) | (-1.567) |
| _cons | -1.677*** | -2.066*** | | -1.646** | -2.868*** | | -1.761 | -2.041 | |
| | (-2.706) | (-3.586) | | (-2.165) | (-4.191) | | (-1.536) | (-1.534) | |
| atanhrho_12 | -0.611*** | | | -0.562** | | | -0.311 | | |
| | (-2.998) | | | (-2.118) | | | (-1.083) | | |
| Observations | 6441 | 6441 | 6441 | 4616 | 4616 | 4616 | 1825 | 1825 | 1825 |
| Province FE | Yes | Yes | Yes | Yes | Yes | Yes | Yes | Yes | Yes |

Note: (1) *, **, and *** represent significance at the 10%, 5%, and 1% levels, respectively. (2) The numbers shown in parentheses are z values under robust standard errors. (3) The first and second stages shown in the table report the regression coefficients of the probit model, and the marginal coefficients report the average marginal effects of the second-stage probit model.

endogeneity bias, the labor force participation rate of women in families with parents increases by 40.3% compared with that of women in families without parents.

The results shown in Tables 9 and 10 suggest that coresidence with parents still significantly promotes women's labor market participation when endogeneity is mitigated and that the probit and OLS models significantly underestimate the promotion degree of coresidence with parents to women's participation in the labor market.

## 5.2 Heterogeneity analysis

This paper analyzes the heterogeneity of the samples based on the age group of the children (for families with multiple children, this paper divides them according to the age of the youngest child in the family), whether the minor children include boys, and whether the husband is

**Table 10. The CMP estimation results of the impact of residential patterns on female labor market participation (*IV_2*).**

| | The national samples | | |
| --- | --- | --- | --- |
| | **First** | **second** | **margin** |
| IV_2 | 0.148** | | |
| | (2.004) | | |
| coresidence | | 1.371*** | 0.403*** |
| | | (8.851) | (7.401) |
| age | 0.062 | 0.104* | 0.031* |
| | (0.992) | (1.667) | (1.690) |
| age_square | -0.099 | -0.090 | -0.027 |
| | (-1.306) | (-1.205) | (-1.216) |
| ethnic | -0.188* | 0.072 | 0.021 |
| | (-1.749) | (0.698) | (0.696) |
| health | 0.074* | -0.107*** | -0.031*** |
| | (1.835) | (-3.004) | (-3.024) |
| edu | -0.023 | 0.118*** | 0.035*** |
| | (-0.969) | (4.921) | (5.227) |
| income_husband | -0.013* | -0.009 | -0.003 |
| | (-1.940) | (-1.321) | (-1.340) |
| children_number | 0.184*** | -0.085 | -0.025 |
| | (2.825) | (-1.382) | (-1.375) |
| boy_number | -0.062 | 0.118* | 0.035* |
| | (-0.839) | (1.781) | (1.793) |
| preschooler | 0.093 | -0.277*** | -0.081*** |
| | (0.957) | (-2.932) | (-2.974) |
| adult_number | 0.381** | -0.185 | -0.055 |
| | (2.479) | (-1.307) | (-1.298) |
| _cons | -1.439 | -2.549** | |
| | (-1.107) | (-2.015) | |
| atanhrho_12 | -1.176*** | | |
| | (-3.111) | | |
| Observations | 1841 | 1841 | 1841 |
| Province FE | Yes | Yes | Yes |

Note: (1) *, **, and *** represent significance at the 10%, 5%, and 1% levels, respectively. (2) The numbers shown in parentheses are z values under robust standard errors. (3) The first and second stages shown in the table report the regression coefficients of the probit model, and the marginal coefficients report the average marginal effects of the second-stage probit model.

employed to explore the heterogeneous effect of residential patterns on women in different families, the findings of which are conducive to targeted policy recommendations.

**5.2.1 The impact of residential patterns on female labor market participation.** Part A shown in Table 11 groups the families according to the age of the children. The regression results show that the coresidential pattern has the most significant contribution to the labor force participation of women in families with children aged 0–6, while there is no significant effect for women in families with children aged 7–12 or 13–18. The differences between the groups are highly significant. Usually, the younger the child is, the heavier the burden of child-care is. For families with children aged 0–6, the grandchild care effect of coresidence with parents has a large spillover effect that significantly promotes women's labor market participation; when the children are more than six years old, they are of school age, and the burden of

childcare for women is reduced. Thus, the spillover effect of coresidence with parents will be reduced accordingly such that the promotion effect of living together on the possibility of female employment participation is no longer significant.

Part B shown in Table 11 groups the families according to whether the minor children include boys. The regression results show that the residential pattern of coresidence with parents significantly promotes female labor market participation in families with and without boys. However, the promotion effect in families without boys is much more significant than that in families with boys. The difference in the coefficients between the groups is significant at the 5% level. The reason may be that Chinese people have a strong preference for boys; thus, married women are more inclined to provide personal care for families with boys, which means that the spillover effect of grandchild care is weakened. For families without boys, grandchild care will greatly relieve women of childcare-related stress, which means that the pattern of coresidence with parents has a stronger effect on promoting female labor force participation in such families.

Part C shown in Table 11 groups the families according to whether the husband is employed. The regression results show that for families in which the husbands are not employed, the promotion effect of coresidence with parents on female labor market participation is no longer significant. For families in which the husbands are employed, the promotion effect of coresidence with parents on female labor market participation remains significant, and the difference is significant at the 10% level. For families in which the husband is not employed, the spillover effect of coresidence with parents is no longer significant because the husband can share the childcare-related pressure; however, for families in which the husband is employed, the spillover effect of coresidence with parents is effective in relieving women's childcare-related pressure and thus significantly promotes the possibility of women participating in the labor market.

**5.2.2 The impact of residential patterns on female labor market income.** Table 12 analyzes the impact of residential patterns on female labor market income according to the same classification criteria. When the sample is divided according to the age group of the children and whether there are boys or not, the empirical p values are no longer significant, indicating that there are no significant differences between the samples. For Part C, the spillover effect of coresidence with parents significantly promotes women's work performance if the husband is employed.

### 5.3 Robustness testing

**5.3.1 Index replacement.** When exploring the impact of residential patterns on female labor market performance, the core explanatory variable is replaced by the number of parents (*coresidence_num*) living together. The regression results are consistent with the benchmark regression results, indicating that the conclusion made earlier in this paper is relatively robust. Specific regression results are shown in S1 Appendix.

**5.3.2 Replace the sample with men.** According to existing research, women are more likely than men to be influenced by traditional cultural concepts and spend more time and energy on family care; men are almost unaffected by these factors [2, 3]. Thus, this paper replaces the research object with men who are married and have minor children and finds that living with parents has no significant effect on men's labor market participation. Instead, it has a negative effect on men's earnings because coresidence with elderly relatives increases the level of stress related to elderly care for men. The regression results show that for families with minor children, the wife remains the primary caregiver, and coresidence with parents is one of

**Table 11. The heterogeneity of the impact of residential patterns on female labor market participation.**

|  |  | coresidence | Z values/T values | Observations | Empirical P values |
|---|---|---|---|---|---|
| **A. The age of the children** |  |  |  |  |  |
| 0–6 years old | Probit | 0.107*** | 4.301 | 1875 |  |
|  | OLS | 0.104*** | 4.320 | 1875 |  |
| 7–12 years old | Probit | 0.023 | 1.046 | 2112 | 0.029 |
|  | OLS | 0.024 | 1.070 | 2112 | 0.018 |
| 13–18 years old | Probit | 0.033 | 1.590 | 2454 | 0.094 |
|  | OLS | 0.031 | 1.564 | 2454 | 0.028 |
| **B. Whether there are boys** |  |  |  |  |  |
| Families without boys | Probit | 0.102*** | 4.258 | 2132 |  |
|  | OLS | 0.100*** | 4.486 | 2132 |  |
| Families with boys | Probit | 0.030* | 1.896 | 4309 | 0.012 |
|  | OLS | 0.030* | 1.924 | 4309 | 0.009 |
| **C. Whether the husband is employed** |  |  |  |  |  |
| Unemployed | Probit | -0.044 | -0.764 | 411 |  |
|  | OLS | -0.044 | -0.691 | 411 |  |
| Employed | Probit | 0.064*** | 4.678 | 6026 | 0.075 |
|  | OLS | 0.063*** | 4.882 | 6026 | 0.093 |

Note: (1) *, **, and *** represent significance at the 10%, 5%, and 1% levels, respectively. (2) The "coresidence" category shown in Table 11 represents the coefficients of *coresidence*. (3) The probit model in the table reports the average marginal effects. (4) The "empirical *p* value" is used to test the significance of the difference in coefficients between groups. This paper uses the Chow test. (5) If the sample is divided into three groups, the "empirical *p* value" shows the difference between the group and the first group of samples.

the ways to effectively share the wife's caregiving-related pressure. Thus, the main conclusions of this paper are robust. Specific regression results are shown in S1 Appendix.

**5.3.3 Change in regional fixed effects.** The influence of traditional concepts on women may vary because cities are not equally developed. This section further refines the province-level fixed effects to city-level fixed effects, and the conclusion remain robust. Specific regression results are shown in S1 Appendix.

**Table 12. Heterogeneity in the impact of residential patterns on female labor market income.**

|  | coresidence | Z values/T values | Observations | Empirical P values |
|---|---|---|---|---|
| **A. The age of the children** |  |  |  |  |
| 0–6 years old | 0.263*** | 3.180 | 900 |  |
| 7–12 years old | -0.002 | -0.020 | 993 | 0.880 |
| 13–18 years old | 0.033 | 0.389 | 1151 | 0.270 |
| **B. Whether there are boys** |  |  |  |  |
| Families without boys | 0.135 | 1.538 | 1049 |  |
| Families with boys | 0.066 | 1.254 | 1995 | 0.435 |
| **C. Whether the husband is employed** |  |  |  |  |
| Unemployed | -0.067 | -0.255 | 194 |  |
| Employed | 0.146*** | 2.958 | 2847 | 0.016 |

Note: (1) *, **, and *** represent significance at the 10%, 5%, and 1% levels, respectively. (2) The "coresidence" category shown in Table 12 represents the coefficients of *coresidence*. (3) The "empirical *p* value" is used to test the significance of the difference in coefficients between groups. This paper uses the Chow test. (4) If the sample is divided into three groups, the "empirical *p* value" shows the difference between the group and the first group of samples. (5) The table reports the coefficients of Heckman's second stage.

## 6. Discussion

The labor market performance of Chinese females has long been the focus of academic attention. Influenced by traditional culture, women tend to devote more time and energy to family affairs, which correspondingly reduces their labor market supply and their accumulation of human capital. For women who are married and have minor children, childcare tends to reduce women's labor market participation and income; thus, and it has become an increasingly common social phenomenon for such families to live with elderly individuals. Previous studies on female labor market performance have mainly focused on the negative impact of elderly care or the promotion effect of grandchild care on female employment. Fewer studies have focused on these two effects at the same time, combining the internal logic of their impacts on female labor market performance. Based on social reality, this paper uses 2019 CHFS data to test the specific impact of coresidence with parents on the labor market performance of married women and determine the internal mechanism, which is of great significance to promoting women's labor market performance and releasing the related economic growth potential.

This study provides a new perspective for promoting female labor market performance. Against the background of the labor supply shortage in China and due to the influence of traditional culture, women still play a major role in childcare, which hinders the performance of the female labor market. Elderly individuals play an important role in childcare, which can effectively relieve women's childcare-related stress and promote their labor market performance. Thus, the government should fully affirm the contributions of elderly relatives and formulate appropriate subsidy policies for multigenerational families. Women may be faced with increased levels of childcare-related stress due to the recent implementation of the three-child policy. However. women cannot rely solely on elderly parents to help relieve their parenting-related pressure by caring for their grandchildren. Strengthening public childcare services is a fundamental means by which to alleviate women's childcare-related pressure and promote their employment. For example, more high-quality childcare institutions should be built. In addition, the state should call on men in married families with children to actively share childcare responsibilities to reduce women's worries about employment.

There are some limitations to the research discussed in this paper. For example, due to the limitations of the questionnaire, this paper cannot obtain the specific situation of parents who are not included in the extended family samples; furthermore, the questionnaire does not directly ask questions related to childcare. Thus, it is not possible to accurately measure the crowding-out effect and the spillover effect, which means that we can only look for related proxy variables. Also, with the rapid growth in the number of elderly people and the increasing life expectancy of the population, the burden of old-age care on families will continue to increase. In the future, social old-age care may gradually become more prevalent and take the place of family old-age care, which would mean that women who relieve their childcare-related pressure by living with elderly relatives may face new challenges. All these factors require more informative and detailed data samples, which constitute important directions for our future research.

## 7. Conclusions

Based on micro-level household survey data, this paper finds that for women who are married and have minor children, living with their parents significantly promotes their labor market participation possibilities and has a positive impact on their income. The mechanism analysis shows that there is grandchild care associated with living with parents, which has a spillover effect on female labor market performance; there is also elderly care associated with living with parents, which has a crowding-out effect on female labor market performance. Since the spillover effect is more significant than the crowding-out effect, living with parents has a positive

impact on women's labor market performance. The heterogeneity analysis shows that in terms of labor force participation rate, the residential pattern has a greater effect on women in families with children aged 0–6, women in families without boys, and women in families with employed husbands. In terms of income, the residential pattern has a more significant impact on women in families with employed husbands.

## Supporting information

**S1 Appendix.**
(DOCX)

## Acknowledgments

We are especially grateful to the China Household Financial Survey and Research Center for the data support.

## Author Contributions

**Conceptualization:** Siyan Zhou.

**Data curation:** Siyan Zhou.

**Formal analysis:** Siyan Zhou.

**Funding acquisition:** Qing Wang.

**Investigation:** Siyan Zhou.

**Methodology:** Siyan Zhou.

**Resources:** Siyan Zhou, Qing Wang.

**Software:** Siyan Zhou.

**Supervision:** Qing Wang.

**Validation:** Siyan Zhou.

**Visualization:** Siyan Zhou.

**Writing – original draft:** Siyan Zhou.

**Writing – review & editing:** Siyan Zhou.

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
