## [Decision Letter · Decision Letter 0]

16 Aug 2023

PONE-D-22-34476Do residential patterns affect women's labor market performance? An empirical study based on the CHFS dataPLOS ONE

Dear Dr. Siyan Zhou,

Thank you for submitting your manuscript to PLOS ONE. After careful consideration, we feel that it has merit but does not fully meet PLOS ONE’s publication criteria as it currently stands. Therefore, we invite you to submit a revised version of the manuscript that addresses the points raised during the review process.

The paper is interesting and worth publishing because it addresses Chinese women's problems coping with family issues when looking for a promotion in the labor market. Although the empirical analysis is robust and comprehensive, it must be clarified and explained. Most tables depict results correctly, but some parts need a deeper explanation. In particular, more detailed descriptive information for the variables included in the analysis must be provided. Similarly, a more detailed description of the role played by ethnicity is needed since the model considers the condition of belonging to the Han ethnic group. Also, clarifications about the statistical approach must be added, such as the justification for using instrumental variables, especially regarding how the sample's self-selection problem has been addressed. These drawbacks can be addressed by renaming the fifth section (Further analysis) as Discussion. It is also necessary to expand the sixth section (Conclusions) to add paragraphs discussing the paper's limitations, the policy implications of the empirical results, and the proposals for future lines of research.

Finally, the paper needs a comprehensive revision of English usage, preferably by a native English speaker. The reference section also needs a complete correction for errors, missing information and irregular citations.

Due to these conditions, the editor’s decision is that authors must revise the whole manuscript.

To this effect, you have 45 days to address these comments, requiring resubmission of the corrected version to Journal PLOS ONE.

I encourage you to attend to the reviewers' comments fully. In particular, I ask you to pay attention to the following issues.

Required changes

Review the manuscript's structure to add a discussion section and to expand conclusions.

State the advantages and limitations of the methodology, especially the self-selection problem and the use of the instrumental variable approach.

Perform a comprehensive revision for typography errors and punctuation.

Provide access to the database.

Clarify and explain all technical terms and define their meaning when necessary.

The bibliography must be homogenized and revised.

We look forward to receiving your revised manuscript.

Kind regards,

Humberto Merritt, PhD

Academic Editor

PLOS ONE

Journal Requirements:

“Supported by National Natural Science Foundation of China (NSFC) (Project No. 7194100222). The project provided only financial support for this study. Meanwhile, the funders had no role in study design, data collection and analysis, decision to publish, or preparation of the manuscript.”

Additional Editor Comments (if provided):

The paper is interesting and worth publishing because it addresses Chinese women's problems coping with family issues when looking for a promotion in the labor market. Although the empirical analysis is robust and comprehensive, it must be clarified and explained. Most tables depict results correctly, but some parts need a deeper explanation. In particular, more detailed descriptive information for the variables included in the analysis must be provided. Similarly, a more detailed description of the role played by ethnicity is needed since the model considers the condition of belonging to the Han ethnic group. Also, clarifications about the statistical approach must be added, such as the justification for using instrumental variables, especially regarding how the sample's self-selection problem has been addressed. These drawbacks can be addressed by renaming the fifth section (Further analysis) as Discussion. It is also necessary to expand the sixth section (Conclusions) to add paragraphs discussing the paper's limitations, the policy implications of the empirical results, and the proposals for future lines of research.

Finally, the paper needs a comprehensive revision of English usage, preferably by a native English speaker. The reference section also needs a complete correction for errors, missing information and irregular citations.

Due to these conditions, the editor’s decision is that authors must revise the whole manuscript.

To this effect, you have 45 days to address these comments, requiring resubmission of the corrected version to Journal PLOS ONE.

I encourage you to attend to the reviewers' comments fully. In particular, I ask you to pay attention to the following issues.

Required changes

Review the manuscript's structure to add a discussion section and to expand conclusions.

State the advantages and limitations of the methodology, especially the self-selection problem and the use of the instrumental variable approach.

Perform a comprehensive revision for typography errors and punctuation.

Provide access to the database.

Clarify and explain all technical terms and define their meaning when necessary.

The bibliography must be homogenized and revised.

Reviewers' comments:

Reviewer's Responses to Questions

**Comments to the Author**

1. Is the manuscript technically sound, and do the data support the conclusions?

Reviewer #1: Yes

Reviewer #2: Partly

2. Has the statistical analysis been performed appropriately and rigorously? 

Reviewer #1: Yes

Reviewer #2: No

3. Have the authors made all data underlying the findings in their manuscript fully available?

Reviewer #1: No

Reviewer #2: Yes

4. Is the manuscript presented in an intelligible fashion and written in standard English?

Reviewer #1: No

Reviewer #2: Yes

5. Review Comments to the Author

Reviewer #1: Introduction: women are especially vulnerable in the labor market not only because of the household responsibilities but also because of the long-standing gender discrimination against women. It would be helpful to briefly introduce the existing fundings on the later aspect. In addition, in terms of how the Chinese culture undermined married women’s labor market participations and income, how was the gender differences in the labor market of other countries, where there is no such culture? It would help readers to understanding the topic of this paper if the authors can extend the existing evidence to other factors that could affect women’s job market participations (like gender discrimination) and compare findings with international studies.

Theoretical basis and research hypothesis: I don’t quite understand the third hypothesis that why the “spillover effect” is dominant. My feeling is that which effects is dominant will depend on the health status of grandparents – whether they are too sick to live independently or whether they are healthy enough to take care of the grandchild. I also don’t understand what “preferring the young over the old” means and how this is related to the comparison between the spillover and crowding out effect. Could you please elaborate the rationale of your third hypothesis.

Variables in Table 1 and Table 2 & Results: There are distinct differences in childrearing workload and requirements among different ages of children, for example, preschoolers require full-day attention whereas middle-school students don’t. In addition, grandparents might be able to feed children, but it might be difficult for them to help with coursework for older-age children. Therefore, children’s age could play an important role, but I did not see it was included in the study. Second, it is surprising to see the women’s income is higher than their husband’s. Could you explain why? Lastly, the results showed heterogenous effects by child gender, but I did not see child’s gender is included in Table 1 or Table 2. Please provide the descriptive information of child age and gender in Table 1 and Table 2, also include them into the regression tables from Table 3 to Table 5.

Discussion: The discussion section is missing. The authors should have a discussion section to explain their results, compare the fundings with literature, specify the implications, limitations, and contributions of the study. It will help readers to understand the value of the current study.

Reviewer #2: This paper addresses an interesting and important research question. The research question is well motivated and there is clear discussion of contributions. However, I am not sure if the identification strategy is able to robustly address the research question. I present my main reservations with the identification strategy below:

1) Is the proportion of households living with parents (excluding observing HH) in the community a good instrument?

The relevance condition of the IV is only met if there is no self-selection. However, it is possible for a family to have internally migrated into a community with high proportion of households where families live with their parents. Could the authors use other dataset to establish a large time lag for the IV so that any concerns of internal migration convoluting the IV is reduced?

In any case, it would be good for the authors to present summary statistics of the IV in Table 2, and also to separately run correlation of the community variable to geographic variables like urban and district.

I am also concerned that the community level IV maybe picking up some unobservables that are correlated with the error term. One is fairly obvious: as evident in Table 3, job opportunities and the need for coresidence may be greater in urban areas. In other words, what may be driving some communities to be traditionnaly more prone to coresidence may the thriving business and work opportunities--which would violate the exclusionary restriction assumption.

2) It is important to control for the number of adults in the households as well. In households with coresidence, if there are also other additional adult members (which is not currently controlled for), coresidence maybe picking up the effect of the childcare provisions of these adult members.

Other salient points:

3) It might be good to have a more critical discussion of the literature and clearly identify why it is difficult to establish causality from coresidence to female labour market participation. It might also be pertinent to broaden the discussion to look at some of the major factors that affect labour market participation (see Klasen et al 2020, for example) across the world/developing world and to then contextualise this to China.

4) What happens to the income_husband if the husband is an entrepeneur? does it take a value of zero or is the profit from the enterprise taken reported in the survey and captured in the data?

5) When interpreting the main results, there is discussion of the sign and significance of the key variables of interest but not the magnitude. It would be good if the authors are able to interpret the main effects with respect to magitude of effects as well.

Minor points:

6) Table 8 onwards could be appendices. In any case the authors need to present notes to each of these tables. In general (including for Tables 3-7), the notes to the tables should be self-sufficient and should provide readers sufficient information to read the table without having to scroll back to the text.

7) The last sentence in the conclusion seems to be not drawn from the results and may be making recommendations beyond the scope of the study. I suggest removing it, unless the authors are back these with further analysis.

References

Klasen, S., Le, T. T. N., Pieters, J., & Santos Silva, M. (2021). What drives female labour force participation? Comparable micro-level evidence from eight developing and emerging economies. The Journal of Development Studies, 57(3), 417-442.

6. PLOS authors have the option to publish the peer review history of their article (what does this mean?). If published, this will include your full peer review and any attached files.

Reviewer #1: **Yes: **Qi Jiang

Reviewer #2: **Yes: **Vengadeshvaran J. Sarma

---

## [Author Response · Author response to Decision Letter 0]

29 Sep 2023

Response to Review Comments

(Manuscript Number: PONE-D-22-34476)

Dear Editor,

Thank you very much for giving us a chance to revise our manuscript. The academic editor and the reviewers’ comments are valuable and very helpful for improving our research paper. We have carefully read all comments and have tried our best to revise the manuscript as their suggestions, which we hope to meet with acceptance requirements. We used a revised model in the original manuscript, named “Revised Manuscript with Track Changes” in this submission. In addition, reviewer # 1 comments have been yellow highlighted, reviewer # 2 comments have been blue highlighted in the manuscript.

Response to the academic editor 

Response: We sincerely appreciate your thorough review of our manuscript and the valuable guidance you provided. In response to your comments and suggestions, we have made the following improvements to the article:

1. Empirical Results: We have enhanced the presentation of empirical results by discussing the significance and signs of core explanatory variables and control variables. We have provided more detailed explanations of their impact and discussed the implications of the results. 

2. Ethnicity Variable: We have included an explanation of the inclusion of the ethnicity variable in our model and provided an interpretation of the regression results related to this variable. 

3. Instrumental Variables and Heckman Two-Step Method: We have expanded our explanation of the use of instrumental variables and the Heckman Two-Step method. Additionally, we have introduced additional instrumental variables to ensure the robustness and reliability of our empirical results. 

4. Structural Adjustments: We have made structural adjustments to the article, including the addition of a new section (Section 6) for discussions. We have also relocated the empirical results based on instrumental variables regression from Section 4.2 ("Endogeneity problem") to Section 5.1 ("Endogeneity problem"). Similarly, we moved the "Mechanism analysis" section from Section 5 to Section 4.2. 

5. Conclusion and Recommendations: We have expanded and revised Section 7 (Conclusion and Recommendations), incorporating policy recommendations, a discussion of the paper's limitations, and suggestions for future research. 

6. Language Editing: We engaged a professional editing service to polish the language of the manuscript, and we have included a certificate verifying this editing. 

7. Formatting and References: We have thoroughly proofread the paper for formatting and reference style to ensure compliance with the required guidelines.

8. Data: Concerning data access, we utilized publicly available datasets, which can be obtained from the official website: https://chfs.swufe.edu.cn. We have also upload data to the public database OPENICPSR, Resource URI: /openicpsr/194102, Please visit the following URL to access the project:

https://deposit.icpsr.umich.edu/deposit/claimResource?tenant=openicpsr&claimId=120703. 

9. Funding statement: Furthermore, we have updated the funding statement as per your recommendation and included it in the cover letter.

Response to Reviewer #1 Comments

I am very much thankful to your deep and thorough review. In response to your comments, we have made changes in the article and highlighted them with yellow shading.

Comment 1: Introduction: women are especially vulnerable in the labor market not only because of the household responsibilities but also because of the long-standing gender discrimination against women. It would be helpful to briefly introduce the existing fundings on the later aspect. In addition, in terms of how the Chinese culture undermined married women’s labor market participations and income, how was the gender differences in the labor market of other countries, where there is no such culture? It would help readers to understanding the topic of this paper if the authors can extend the existing evidence to other factors that could affect women’s job market participations (like gender discrimination) and compare findings with international studies.

Response: Thank you for your suggestions regarding the "Introduction" section. Women in the labor force are both important and unique, important because they constitute a significant portion of the labor force, and unique because they are often influenced by societal traditional norms and gender discrimination, which hinder their performance in the labor market. On one hand, traditional norms directly affect women, reducing their labor force participation rates. On the other hand, these traditional norms can also lead to and exacerbate societal discrimination against women. Under the influence of traditional norms, many people may hold negative views about women engaging in paid work, making it more difficult for women to find employment and reducing their willingness to work. Therefore, your input on gender discrimination greatly enriches the article's perspective and enhances its internal logic. We have made additional explanations in the "Introduction" section to address this.

Furthermore, the phenomenon of traditional culture and gender discrimination hindering women's employment is not unique to China; in fact, it exists in many countries worldwide, both developed and developing. We have also added relevant literature to the “Theoretical basis and research hypothesis” section.

Comment 2: Theoretical basis and research hypothesis: I don’t quite understand the third hypothesis that why the “spillover effect” is dominant. My feeling is that which effects is dominant will depend on the health status of grandparents – whether they are too sick to live independently or whether they are healthy enough to take care of the grandchild. I also don’t understand what “preferring the young over the old” means and how this is related to the comparison between the spillover and crowding out effect. Could you please elaborate the rationale of your third hypothesis.

Response: For multi-generational families, there are primarily two forms of intergenerational relationships within the family: one is the support relationship from the younger generation to the older generation, and the other is the nurturing relationship from the older generation to the younger generation. In multi-generational families, these family dynamics often revolve around the middle-aged couples, and with the influence of economic development and social policies, intergenerational relationships within the family are also changing. Regarding the relationship between middle-aged couples and their parents, middle-aged couples have grown up, married, and had children, and the nurturing relationship from the older generation to the younger generation no longer exists. As for the support relationship from the younger generation to the older generation, traditional Chinese culture emphasizes "raising children for old-age support," meaning that adult children should take on the responsibility of supporting and caring for their parents in their old age, providing financial assistance and daily care for their elderly parents. However, with economic development and the gradual improvement of social security, the elderly in China no longer primarily require financial support from their children but rather focus on emotional needs. In multi-generational families, when the elderly are in good health, they may even provide assistance to their own children, including financial support and helping middle-aged couples take care of their grandchildren. Regarding the relationship between middle-aged couples and their children, since their children are not yet adults, there is no nurturing relationship from the younger generation to the older generation at this stage. In terms of the nurturing relationship from the older generation to the younger generation, the implementation of the one-child policy has artificially reduced the number of children in each family, elevating the status of children within the family rapidly. The degree of attention that middle-aged couples pay to their young children continues to deepen (while at the same time, the level of attention that the elderly pay to their grandchildren also deepens). In this context, middle-aged couples often devote more energy and financial resources to raising the next generation rather than supporting the previous generation. Therefore, in the context of the new elderly care pattern and family planning policies, on the one hand, middle-aged couples reduce their care for the elderly, and it is more often the elderly who assist their own children, such as helping their children take care of underage grandchildren. On the other hand, middle-aged couples invest more energy and money into raising their own children. This has led to a new pattern of Chinese family relationships known as “preferring the young over the old.”

Influenced by traditional culture, women tend to take on a greater share of household caregiving responsibilities. In multigenerational households, middle-aged couples often experience a "care for the elderly" dynamic in their relationship with their parents, which increases the caregiving burden on women and results in a crowding-out effect on their employment. As the economy develops and social security gradually improves, the elderly's retirement needs are changing, and they may assist their adult children, for example, by helping middle-aged children take care of their minor grandchildren, a phenomenon known as "intergenerational caregiving." This can lead to a spillover effect on women's employment. Under the implementation of new elderly care patterns and family planning policies, the traditional concept of "raising children for old-age support" is gradually weakening. It often becomes the case that parents provide support to their children rather than the other way around, reducing the crowding-out effect. In such families, minor grandchildren often hold a high position within the family structure. From the perspective of assisting their children and caring for their grandchildren, elderly individuals may help their middle-aged children take care of their grandchildren, thereby reducing women's burden of caring for young children and enhancing the spillover effect. In this context, the spillover effect is greater than the crowding-out effect. As a result, in multigenerational households where middle-aged couples live with their parents, women's labor market performance tends to improve. This leads us to hypothesis 3.

In the mechanism testing section, we provide evidence for the existence of the crowding-out effect generated by elderly care and the spillover effect generated by cross-generational caregiving. With both effects coexisting, the spillover effect dominates. Therefore, for married couples with children, living with the elderly promotes women's labor market performance. 

Thank you for your valuable input, which has improved the article's logic. We have also made relevant additions in the hypothesis testing section 2.3.

Comment 3: Variables in Table 1 and Table 2 & Results: There are distinct differences in childrearing workload and requirements among different ages of children, for example, preschoolers require full-day attention whereas middle-school students don’t. In addition, grandparents might be able to feed children, but it might be difficult for them to help with coursework for older-age children. Therefore, children’s age could play an important role, but I did not see it was included in the study. Second, it is surprising to see the women’s income is higher than their husband’s. Could you explain why? Lastly, the results showed heterogenous effects by child gender, but I did not see child’s gender is included in Table 1 or Table 2. Please provide the descriptive information of child age and gender in Table 1 and Table 2, also include them into the regression tables from Table 3 to Table 5.

Response: For children of different age groups, their caregiving needs may vary, so the age of the children plays a crucial role and needs further control. Additionally, the gender of the children may also affect whether women participate in the labor market and their income. We appreciate the reviewer's suggestions and reminders, and we have added whether there are boys (boy_number) and whether there are preschool children (preschooler) as control variables, incorporating them into the subsequent regression analysis. We have also included explanations and descriptive statistics for these two variables in Tables 1 and 2.

In the descriptive statistics, women's average income is higher than that of their husbands, which is due to the different statistical criteria for the two variables. Women's income is only used in the second-stage regression because we employ the Heckman two-step method. Therefore, this variable is based on the income of women who participate in the labor market, and samples of women who do not participate in the labor market are not included. In contrast, men's income (income_husband) is a control variable when explaining whether women participate in the labor market (work), and it is also a control variable in the first-stage regression of the Heckman two-step method. Consequently, all samples, whether men participate in the labor market or not, are included in this variable's statistical analysis. We assign a value of 0 to samples where men do not participate in the labor market. As a result, the statistical results show that women's average income is higher than men's.

Comment 4: Discussion: The discussion section is missing. The authors should have a discussion section to explain their results, compare the fundings with literature, specify the implications, limitations, and contributions of the study. It will help readers to understand the value of the current study.

Response: Thank you for your suggestions. Following your advice, we have added a sixth section, the discussion section. In the discussion section, we have made the following improvement: (1) provide explanations for the results of the entire article;(2) compare them with the existing literature, highlight the research's significance; (3) add policy recommendations; (4) address the article's limitations; (5) suggest directions for future research.

Response to Reviewer #2 Comments

I am very much thankful to your deep and thorough review. In response to your comments, we have made changes in the article and highlighted them with blue shading.

Comment 1: Is the proportion of households living with parents (excluding observing HH) in the community a good instrument? The relevance condition of the IV is only met if there is no self-selection. However, it is possible for a family to have internally migrated into a community with high proportion of households where families live with their parents. Could the authors use other dataset to establish a large time lag for the IV so that any concerns of internal migration convoluting the IV is reduced? In any case, it would be good for the authors to present summary statistics of the IV in Table 2, and also to separately run correlation of the community variable to geographic variables like urban and district. 

I am also concerned that the community level IV maybe picking up some unobservables that are correlated with the error term. One is fairly obvious: as evident in Table 3, job opportunities and the need for coresidence may be greater in urban areas. In other words, what may be driving some communities to be traditionnaly more prone to coresidence may the thriving business and work opportunities--which would violate the exclusionary restriction assumption.

Response: Thank you for your relevant suggestion. First, regarding the issue of family migration, a family might move to a county with a higher proportion of living with parents, affecting the relevance of the instrumental variable. According to the CHFS 2019 questionnaire, only 4.56% of families had engaged in cross-county migration, so the proportion of family migration is small and unlikely to affect the relevance of the instrumental variable. Even after excluding samples with migration behavior, according to the regression results, the instrumental variable remains highly correlated with the core explanatory variable. We have also placed the regression results using the instrumental variable after excluding samples with migration behavior in the appendix.

Secondly, regarding the concern about whether the instrumental variable is correlated with the error term, the underlying logic of selecting the proportion of families living with parents in the same county as the instrumental variable is that regions with a stronger traditional concept of "raising children for old-support" are more likely to live with their parents, but this traditional concept does not affect labor market participation. According to the regression results in Table 3, living with parents has a positive effect on female labor market participation, but the living-with-parents effect of one family does not spill over to another family, meaning that the living-with-parents effect of family A does not promote the female labor market participation of family B. When calculating this instrumental variable, we also exclude the current family from the sample calculation, so family A is not included in the sample calculation when calculating the instrumental variable value for family A. Therefore, this instrumental variable meets the condition of being uncorrelated with the error term.

Additionally, you mentioned that prosperous business and job opportunities in urban areas might lead to a higher probability of living with parents. In fact, according to the 2019 CHFS survey data, the proportion of married rural families living with parents is 23.42%, while urban families have a proportion of 14.99%. For married rural families with underage children living with parents, the proportion is 31.07%, while urban families have a proportion of 17.70%. The probability of living with parents in rural areas is higher than in urban areas, but the promoting effect of living with parents on female employment is more pronounced in urban areas. Therefore, this instrumental variable also meets the condition of being uncorrelated with the error term.

Based on your concerns about the instrumental variable, in this revision, we have added a new instrumental variable to mitigate endogeneity and supplemented it in Sections 3.1, 3.2 and 5.1. We constructed the second instrumental variable (IV_2) based on a direct question in the questionnaire that measures whether the household head holds the traditional concept of "raising children for old-support." If the household head holds this traditional concept, the variable takes a value of 1; otherwise, it takes a value of 0. Since the household head is the decision-maker for family affairs, their beliefs will influence the decision of whether the family lives with parents, so we examined whether the household head holds the traditional concept of "supporting the elderly with children" and used it as an instrumental variable. If the household head holds this concept, the family is more likely to live with the elderly for caretaking and support, but the household head's belief in "raising children for old-support" is not directly related to the employment decisions of family females. Therefore, this variable meets the requirements of being directly related to the endogenous variable and uncorrelated with the disturbance term. The 2015 Chinese Household Finance Survey (CHFS) questionnaire measured whether the household head holds the traditional concept of "raising children for old-support" with the following question: "What do you think is the main reason for raising children? 1. To carry on the family line; 2. Affection for children based on emotional considerations; 3. To support the elderly with children; 4. To maintain marital stability; 5. Other." If the household head's choice includes the third option, then it is considered that the household head holds the traditional concept of "raising children for old-support." The CHFS conducts surveys every two years, including some tracking samples and new samples, with questionnaire settings differing from previous years. Since the 2019 questionnaire did not include questions related to "raising children for old-support," and the 2015 questionnaire did not directly measure whether family members live together, we matched the data from 2015 with the data from 2019 using the unique ID of each family, obtaining a total of 1,841 data points for further endogeneity analysis.

Comment 2: It is important to control for the number of adults in the households as well. In households with coresidence, if there are also other additional adult members (which is not currently controlled for), coresidence maybe picking up the effect of the childcare provisions of these adult members.

Response: Thank you for your suggestions. It is possible that having other adults living in the household could also influence childcare, thus affecting the regression results. Therefore, we have included the number of adults in the household (adult_number), excluding the head of the household and the head of the household's parents, as a control variable. We have added explanations and descriptive statistics for this variable in Tables 1 and 2, and it has been incorporated into subsequent regression analyses.

Comment 3: It might be good to have a more critical discussion of the literature and clearly identify why it is difficult to establish causality from coresidence to female labour market participation. It might also be pertinent to broaden the discussion to look at some of the major factors that affect labour market participation (see Klasen et al 2020, for example) across the world/developing world and to then contextualise this to China.

Response: Thank you for your valuable feedback. In contemporary society, the phenomenon of middle-aged couples with young children co-residing with their parents is becoming increasingly common. However, there is currently limited research in the economics literature on this social phenomenon. Furthermore, within this social phenomenon, both "intergenerational caregiving" and "elderly caregiving" factors coexist. Most literature tends to focus solely on the negative impact of elderly caregiving responsibilities on female employment or the promoting effect of intergenerational caregiving on female employment. There is a lack of literature that simultaneously addresses both of these effects, making it challenging to establish a causal relationship between co-residence and women's labor market participation. Therefore, our paper initiates a discussion on the co-residence phenomenon and provides an analytical explanation of the underlying mechanisms from the perspectives of "intergenerational caregiving" and "elderly caregiving." In response to your suggestion, we have also supplemented the relevant literature, enriching the content of the article and improving its overall logic.

The development of labor markets in various countries around the world is of great concern. Thus, in our paper, it is essential to introduce and discuss this current scenario, especially in comparison with the situation in China. Based on your suggestions, we have introduced a comparison of the factors affecting women's participation in the labor market and added relevant literature to enrich the article. Please see “2. Theoretical basis and research hypothesis” for details.

Comment 4: What happens to the income_husband if the husband is an entrepeneur? does it take a value of zero or is the profit from the enterprise taken reported in the survey and captured in the data?

Response: Thank you for your question. Based on the questionnaire's design, income is composed of multiple components, including wage income, bonus income, cash benefits, in-kind benefits, reimbursement expenses, and more. In previous versions, our definition of income only included wage income. In this revision, we expanded the scope of income composition to include bonus income, cash benefits, and others within the statistical framework. The change in the statistical framework did not affect the regression conclusions and demonstrated robust results.

If the husband is an entrepreneur, we define his income as post-tax income, which means income after deducting all taxes, including wages, dividends, profit sharing, and other income received from the company. After preliminary data processing, out of 6441 samples, there were 6 samples where the income of entrepreneur husbands was negative. Since we need to log-transform income, we assigned a value of 0 to these 6 samples, which did not affect the conclusions.

Comment 5: When interpreting the main results, there is discussion of the sign and significance of the key variables of interest but not the magnitude. It would be good if the authors are able to interpret the main effects with respect to magitude of effects as well.

Response: Thank you for your suggestion. In this revision, we have added discussions regarding the magnitude of the main effects. Additionally, we have included discussions about the signs, significance, and magnitude of the control variables, along with explanations of the regression results. The specific additions are located in 4.1.1, 4.1.2 and 5.1. 

Comment 6: Table 8 onwards could be appendices. In any case the authors need to present notes to each of these tables. In general (including for Tables 3-7), the notes to the tables should be self-sufficient and should provide readers sufficient information to read the table without having to scroll back to the text.

Response: Thank you for your suggestion. In this revision, we have placed robustness tests and subsequent tables in the appendix. Furthermore, we have provided annotations for each table, making it unnecessary for readers to scroll back to the main text to understand them.

Comment 7: The last sentence in the conclusion seems to be not drawn from the results and may be making recommendations beyond the scope of the study. I suggest removing it, unless the authors are back these with further analysis.

Response: Thank you for your feedback. Indeed, the last sentence in the conclusion appeared to lack empirical support and went beyond the scope of the study. Therefore, we have removed that sentence. Additionally, we have added a sixth section to provide more detailed content, including policy recommendations, study limitations, and future prospects, to make the article more comprehensive.

---

## [Editor Report · Decision Letter 1]

30 Oct 2023

PONE-D-22-34476R1Do residential patterns affect women's labor market performance?An empirical study based on CHFS dataPLOS ONE

Dear Dr. Zhou,

Thank you for submitting your manuscript to PLOS ONE. After careful consideration, we feel that it has merit but does not fully meet PLOS ONE’s publication criteria as it currently stands. Therefore, we invite you to submit a revised version of the manuscript that addresses the points raised during the review process.

I confirm that the manuscript titled “Do residential patterns affect women's labor market performance? An empirical study based on CHFS data” fulfills the academic and quality criteria for publication in PLOS ONE. Nevertheless, there are two additional requirements to be satisfied before the publication can proceed. One is that research published in PLOS ONE must have been conducted to the highest ethical standards. Because your study does involve survey-based research, you must provide evidence that you have received approval from the relevant responsible body. If approval was not obtained, you must explain why it was not required (for further details, see the criteria for publication guidelines at https://journals.plos.org/plosone/s/criteria-for-publication). The other is a kind suggestion to add the keyword "female labor" to the appropriate entries and to change the keyword "labor market participation" for “labor market inclusion.” You are encouraged to re-submit your manuscript as soon as you have acted on these requirements.

We look forward to receiving your revised manuscript.

Kind regards,

Humberto Merritt, PhD

Academic Editor

PLOS ONE

Journal Requirements:

Additional Editor Comments :

After carefully revising the recently submitted version of the manuscript titled “Do residential patterns affect women's labor market performance? An empirical study based on CHFS data” I confirm that the manuscript now fulfills the academic and quality criteria for publication in PLOS ONE. Nevertheless, there are two additional requirements to be satisfied before the publication can proceed. One is that research published in PLOS ONE must have been conducted to the highest ethical standards. Because your study does involve survey-based research, you must provide evidence that you have received approval from the relevant responsible body. If approval was not obtained, you must explain why it was not required (for further details, see the criteria for publication guidelines at https://journals.plos.org/plosone/s/criteria-for-publication). The other is a kind suggestion to add the keyword "female labor" to the appropriate entries and to change the keyword "labor market participation" for “labor market inclusion.” You are encouraged to re-submit your manuscript as soon as you have acted on these requirements.

---

## [Author Response · Author response to Decision Letter 1]

1 Nov 2023

Dear Editor,

We would like to express our gratitude to you and the reviewers for investing your valuable time in reviewing our paper and for providing insightful comments. It was these insightful remarks that allowed us to make potential improvements in the current version. The authors have taken great care to consider each comment and have made every effort to address them. We hope that the revised manuscript now meets your high standards. We would welcome any further constructive comments. Below, we have provided point-by-point responses, with all modifications in the manuscript highlighted in yellow.

Response to the reviewers’ comments:

Comment 1: Research published in PLOS ONE must have been conducted to the highest ethical standards. Because your study does involve survey-based research, you must provide evidence that you have received approval from the relevant responsible body. If approval was not obtained, you must explain why it was not required (for further details, see the criteria for publication guidelines at https://journals.plos.org/plosone/s/criteria-for-publication).

Response: The data utilized in this paper is derived from the China Household Financial Survey (CHFS), which is managed and organized by the Survey and Research Center for China Household Finance at the Southwestern University of Finance and Economics. The CHFS is a nationwide sample survey designed to gather relevant information about household finance at the micro level, offering a comprehensive and detailed depiction of household economic and financial behaviors. The publicly available data from 2011 to 2019 can be accessed by the public through a designated website for academic research, free of charge. The survey was conducted with the informed consent of each household member. In the publicly available data, personal information is anonymized, and each family and individual is assigned a unique identifier. Consequently, the survey does not involve human experimentation and does not disclose any private information about individuals or families. This paper exclusively employs the data from this research for academic research and non-commercial purposes. As such, ethical approval is not required.

Comment 2: The other requirement is a kind suggestion to add the keyword "female labor" to the appropriate entries and to change the keyword "labor market participation" for “labor market inclusion.” 

Response: Thank you for your suggestion. We have included "female labor" in the list of keywords and made the change from "labor market participation" to "labor market inclusion," as highlighted in yellow in the manuscript.

Comment 3: Please review your reference list to ensure that it is complete and correct. If you have cited papers that have been retracted, please include the rationale for doing so in the manuscript text, or remove these references and replace them with relevant current references. Any changes to the reference list should be mentioned in the rebuttal letter that accompanies your revised manuscript. If you need to cite a retracted article, indicate the article’s retracted status in the References list and also include a citation and full reference for the retraction notice.

Response: Thank you for your advice. We have thoroughly reviewed the references once more and made the following improvements: ensuring that page numbers for relevant documents are complete, correcting any spelling errors, and carefully proofreading author names. The altered references have been marked with yellow highlighting. Upon this review, we have confirmed that we did not cite any literature that has been retracted.

---

## [Editor Report · Decision Letter 2]

6 Nov 2023

Do residential patterns affect women's labor market performance?An empirical study based on CHFS data

PONE-D-22-34476R2

Dear Dr. Zhou,

We’re pleased to inform you that your manuscript has been judged scientifically suitable for publication and will be formally accepted for publication once it meets all outstanding technical requirements.

Kind regards,

Humberto Merritt, PhD

Academic Editor

PLOS ONE

Additional Editor Comments (optional):

After carefully revising the recently submitted revised version of the manuscript titled “Do residential patterns affect women's labor market performance? An empirical study based on CHFS data,” I do confirm that the manuscript now fulfills the academic and quality criteria for publication in PLOS ONE.

So, please follow the submission instructions provided by PLOS ONE for further editorial directions.

Reviewers' comments:

I do confirm that the manuscript now fulfills the academic and quality criteria for publication in PLOS ONE

---

## [Editor Report · Acceptance letter]

9 Nov 2023

PONE-D-22-34476R2 

Do residential patterns affect women's labor market performance? An empirical study based on CHFS data 

Dear Dr. Zhou:

I'm pleased to inform you that your manuscript has been deemed suitable for publication in PLOS ONE. Congratulations! Your manuscript is now with our production department. 

Kind regards, 

on behalf of

Dr. Humberto Merritt 

Academic Editor

PLOS ONE